# Monoterpenoid aryl hydrocarbon receptor allosteric antagonists protect against ultraviolet skin damage in female mice

Karolína Ondrová[1], Iveta Zůvalová [1], Barbora Vyhlídalová [1], Kristýna Krasulová [1], Eva Miková[1], Radim Vrzal[1], Petr Nádvorník[1], Binod Nepal[2], Sandhya Kortagere[2], Martina Kopečná [3], David Kopečný [3], Marek Šebela [4], Fraydoon Rastinejad [5], Hua Pu[5], Miroslav Soural [6], Katharina Maria Rolfes[7], Thomas Haarmann-Stemmann[7], Hao Li[8], Sridhar Mani [8] ✉ & Zdeněk Dvořák [1] ✉

The human aryl hydrocarbon receptor (AhR) is a ligand-activated transcription factor that is a pivotal regulator of human physiology and pathophysiology. Allosteric inhibition of AhR was previously thought to be untenable. Here, we identify carvones as noncompetitive, insurmountable antagonists of AhR and characterize the structural and functional consequences of their binding. Carvones do not displace radiolabeled ligands from binding to AhR but instead bind allosterically within the bHLH/PAS-A region of AhR. Carvones do not influence the translocation of ligand-activated AhR into the nucleus but inhibit the heterodimerization of AhR with its canonical partner ARNT and subsequent binding of AhR to the promoter of *CYP1A1*. As a proof of concept, we demonstrate physiologically relevant Ahr-antagonism by carvones in vivo in female mice. These substances establish the molecular basis for selective targeting of AhR regardless of the type of ligand(s) present and provide opportunities for the treatment of disease processes modified by AhR.

The aryl hydrocarbon receptor (AhR) is a ligand-activated transcription factor that belongs to the family of basic helix-loop-helix transcription factors. In the resting state, unliganded AhR resides in the cytosol. Upon ligand binding to AhR, the ligand-receptor complex translocates to the cell nucleus. It forms a heterodimer with AhR nuclear translocator (ARNT), which binds to specific response elements in the promoters of the target genes. Typical xenobiotic ligands of AhR are environmental contaminants, such as polyaromatic hydrocarbons (e.g., benzo[*a*]pyrene - BaP) and halogenated aromatic hydrocarbons (e.g., 2,3,7,8-tetrachlorodibenzo-*p*-dioxin - TCDD), as well as naturally occurring chemicals, such as various polyphenols. The endogenous ligands of AhR are mainly intermediary and microbial metabolites of tryptophan, such as 6-formylindolo[3,2-*b*]carbazole (FICZ)[1]. AhR regulates the expression of genes involved in xenoprotection, immune response, cell cycle, differentiation, lipid, and carbohydrate metabolism. Therefore, AhR is a pivotal determinant not only in human physiology (e.g., hematopoietic development)[2] but also in the incidence, onset, and progression of pathophysiological processes, including carcinogenesis, inflammation, infection, diabetes, and cardiovascular diseases[3,4]. Most AhR ligands are partial agonists.

[1]Department of Cell Biology and Genetics, Faculty of Science, Palacký University, Olomouc, Czech Republic. [2]Department of Microbiology & Immunology, Drexel University College of Medicine, Philadelphia, PA, USA. [3]Department of Experimental Biology, Faculty of Science, Palacký University, Olomouc, Czech Republic. [4]Department of Biochemistry, Faculty of Science, Palacký University, Olomouc, Czech Republic. [5]Target Discovery Institute Nuffield Department of Medicine Research Building Brasenose College University of Oxford, Oxford, UK. [6]Department of Organic Chemistry, Faculty of Science, Palacký University, Olomouc, Czech Republic. [7]IUF-Leibniz-Research Institute for Environmental Medicine, Düsseldorf, Germany. [8]Department of Medicine, Oncology, Molecular Pharmacology, and Genetics, Albert Einstein College of Medicine, Bronx, NY, USA. ✉e-mail: sridhar.mani@einsteinmed.edu; moulin@email.cz

Full agonists of AhR include halogenated aromatic hydrocarbons, such as TCDD, whereas AhR antagonists are scarce. For instance, the stilbenoid resveratrol or synthetic inhibitor of c-Jun-N-terminal kinase SP600125 were long deemed AhR antagonists until their minimal residual agonist activity was unveiled[5]. The first identified and bona fide frequently used competitive antagonist of AhR was 3′-methoxy-4′-nitroflavone (MNF)[6]; however, AhR-dependent enhancement of *CYP1A1* transcription by MNF was also reported[7]. By screening a 10 K chemical library, 2-methyl-2*H*-pyrazole-3-carboxylic acid (2-methyl-4-*o*-tolylazo-phenyl)-amide (CH223191) was identified as a ligand and potent competitive antagonist ($IC_{50}$ ~ 30 nM) of AhR[8]. Later, a series of CH223191-based antagonists were developed, and the AhR-independent pro-proliferative properties of CH223191 were reported[9]. Additionally, CH223191 is a ligand-selective antagonist of AhR that preferentially inhibits the halogenated aromatic hydrocarbon class of agonists (e.g., TCDD) but not others, such as polyaromatic hydrocarbons or flavonoids[10]. Perdew's lab reported *N*-(2-(1*H*-indol-3-yl)ethyl)-9-isopropyl-2-(5-methyl pyridine-3-yl)-9*H*-purin-6-amine (GNF351) as a high-affinity ligand ($IC_{50}$ ~ 62 nM) and competitive antagonist of AhR with the capability to inhibit both genomic and nongenomic activities of AhR[11,12]. There are isolated reports on the in vitro and in vivo effects of FDA-approved drugs with AhR-antagonist activity. For instance, clofazimine, an antileprosy drug and AhR antagonist, suppressed multiple myeloma in transgenic mice; however, the putative AhR-dependent mechanism was not directly evidenced[13]. Another example is relapse during melanoma treatment with the BRAF inhibitor vemurafenib, which was suggested to be delayed by targeting constitutively active AhR in persisting cells with antagonists[14]. Moreover, we identified vemurafenib as a competitive antagonist of AhR that was found to inhibit the in vitro and in vivo effects of AhR-dependent processes, including the abrogation of anti-inflammatory signaling and response[15]. Currently, two AhR antagonists, BAY2416964 and IK-175, have entered phase 1 clinical trials in order to assess the tolerability and toxicity of the AhR-targeting agents in patients suffering from uncurable solid cancers[16]. Interestingly, AhR was identified as a host factor for Zika and dengue viruses, and the inhibition of AhR boosted antiviral immunity and diminished the in vivo replication of these viruses[17]. Recently, it was demonstrated that AhR is activated by infection with different coronaviruses and that pharmacological inhibition of AhR suppresses the in vivo replication of the viruses HCoV-229 and SARS-CoV-2, the causative agents of the common cold and COVID-19[18].

We reported that the essential oils of dill, caraway, and spearmint have antagonist effects on AhR and that carvones, which are the major constituents of these oils, are responsible for AhR antagonism[19]. Carvone is a monocyclic monoterpenoid that exists in two enantiomers: R-carvone has a sweetish, spearmint-like odor, and S-carvone has a spicy, caraway-like odor. Human exposure to carvones occurs mainly through the dietary intake of carvone-containing foods and beverages[19] and via percutaneous absorption because carvones are used as skin permeabilizers in transdermal patches[20] and various skincare products (hair shampoos, foaming baths, body lotions, liquid soaps, foot powders, etc.).

In this study, we have significantly advanced our understanding of the mechanism by which carvones antagonize human AhR. Biochemical and cell biological studies demonstrated a specific, selective, and efficacious insurmountable and noncompetitive mechanism of AhR antagonism involving the disruption of AhR-ARNT dimerization by carvones in the cell nucleus. As a proof of concept, we showed in vivo that carvones antagonize ligand-inducible expression of Ahr target genes in mouse skin, and reverse the modulation of UV-induced skin inflammatory mediators by Ahr ligands. Collectively, we report small dietary monocyclic monoterpenoids as negative allosteric modulators of AhR with potential preventive and therapeutic applications.

## Results

### Carvones are noncompetitive antagonists of AhR

The human stably transfected reporter cell line AZ-AHR[21] was used to investigate the effects of carvones on the transcriptional activity of AhR. Model full agonists of AhR, including TCDD, BaP, FICZ, indirubin, indole, and 3-methylindole (3-MI), caused a concentration-dependent increase in AhR-mediated luciferase activity (Fig. S1A). Carvones did not affect the basal transcriptional activity of AhR (Fig. S1B). The antagonist effects of carvones and CH223191 (orthosteric AhR antagonist) on agonist-inducible AhR activity were examined in cells co-incubated for 4 h and 24 h with a fixed concentration of agonist ligands (corresponding to their $EC_{80}$) and increasing concentrations of carvones and CH223191. After 4 h of incubation, carvones exerted concentration-dependent antagonist effects on AhR activation by all tested agonists, and the inhibitor constants $K_i$ against all AhR agonists laid in very narrow range, which spanned from 11 μM to 48 μM. On the other hand, CH223191 was unable to inhibit AhR activation by indirubin. The $K_i$ values for CH223191 were much lower than those for carvones, but they highly varied between different agonists (from 0.1 μM to 1 μM) (Fig. 1A). After 24 h of incubation, the antagonist effects of CH223191 against TCDD-activated AhR were much stronger as compared to carvones. Both carvones and CH223191 antagonized BaP-activated AhR to the similar extent. Neither carvones nor CH223191 inhibited FICZ-activated AhR, and both antagonists potentiated the activation of AhR by indirubin (Fig. 1A). These data support the known fact that CH223191 is a ligand-selective antagonist[10], whereas carvones behave as AhR pan-antagonist. Next, we analyzed the mechanism of AhR antagonism. For this purpose, we incubated cells with fixed concentrations of carvones and CH223191, combined with increasing concentrations of AhR agonists. We observed a gradual decrease in $E_{MAX}$ and a slight decline in $EC_{50}$ with increasing concentrations of carvones for each tested agonist at both times of incubation (Fig. 1B, Figure S1C). The exception was indirubin, which was not antagonized by carvones, consistently with results in Fig. 1A. These data imply that carvones are primarily either (insurmountable) noncompetitive, irreversible competitive, or uncompetitive antagonists of AhR. Since the same concentration of carvones antagonized both high and low concentrations of all agonists used to a comparable degree (Table S2), the uncompetitive mechanism was ruled out[22]. We also excluded inhibition of luciferase catalytic activity (Figure S1D) and cytotoxicity (Figure S1E), as culprits of inhibition of AhR activity by carvones. The orthosteric ligand of AhR, CH223191, displayed competitive antagonist behavior against TCDD-activated AhR, as it caused typical rightward shift of sigmoid curves (Fig. 1B). The antagonist effects of CH223191 against other AhR agonists were ambiguous. Of note, whereas CH223191 displayed much lower $K_i$ values than carvones, the relative strength of antagonism is hereby manifested against the agonists at their $EC_{80}$ concentration (Fig. 1A). Carvones were much stronger AhR antagonists as compared to CH223191, when applied against high concentrations of agonists (~$100 \times EC_{80}$) (Fig. 1B).

Finally, we have examined AhR-antagonist activities of different cyclic monoterpenoids as the carvones structural analogs. We found that (+)-dihydrocarvone, R/S-pulegones, piperitone, piperitenone, carvacrol and thymol dose-dependently inhibited TCDD- and BaP-activated AhR. On the other hand (+)/(−)-menthones, (+)/(−)-isomenthones, cuminol, cuminal and *p*-cymene did not display AhR antagonist activity (Figure S2).

### Carvones downregulate AhR target genes

The effects of carvones on the ligand-inducible expression of the prototypical AhR target gene *CYP1A1*[3] were examined in a complementary set of AhR-competent human cell lines, including primary human hepatocytes, HepG2 hepatocarcinoma cells, LS180 colon adenocarcinoma cells, and HaCaT immortal keratinocytes[23]. Carvones inhibited the ligand-inducible mRNA and protein expression of CYP1A1 in cell type-specific and ligand-selective manner. Induction of CYP1A1 by TCDD was

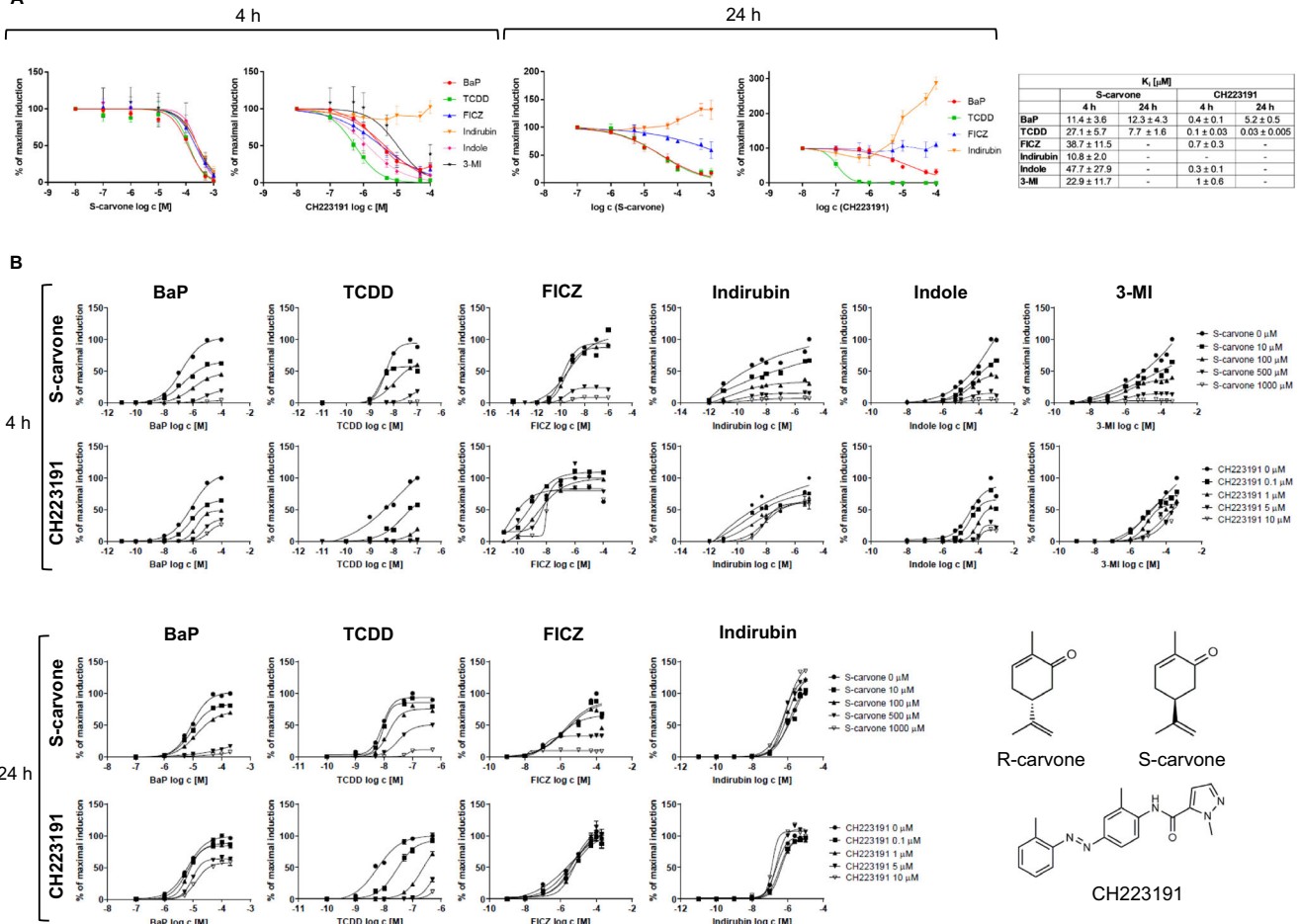

**Fig. 1 | Antagonism of AhR by carvones and CH223191.** A reporter gene assay was carried out in stably transfected AZ-AHR cells incubated for 4 h and 24 h with the tested compounds. Incubations and measurements were performed in quadruplicate (technical replicates). **(A)** Agonist-inducible AhR activity: combined incubations with fixed concentrations of agonists (at $EC_{80}$) and increasing concentrations of carvones and CH223191 ($K_i$ values indicated in the graph). Data are shown as mean ± SD (n = 3). **(B)** Analyses of antagonism mechanism: combined incubations with a fixed concentration of antagonists (carvones and CH223191) and increasing concentrations of AhR agonists. Experiments were performed in three independent cell passages (n = 3). Representative plots from one experiment are shown. Abbreviations: BaP, benzo[a]pyrene; FICZ, 6-formylindolo[3,2-b]carbazole; 3-MI, 3-methylindole; TCDD, 2,3,7,8-tetrachlorodibenzo-p-dioxin. Source data are provided as a Source Data file.

inhibited by carvones in all cell types, by BaP in LS180, HaCaT, and hepatocytes, and by FICZ in HaCaT cells and human hepatocytes (Fig. 2, Figure S3). In addition, TCDD-inducible, AhR receptor-regulated 7-ethoxyresorufin-O-deethylase (EROD) catalytic activity[3] was strongly decreased by S-carvone in AZ-AHR cells (Figure S4A). Carvones also downregulated AhR ligand-inducible canonical genes *CYP1A2* (Figure S3) and *AhRR* (AhR repressor) (Figure S4B) and the noncanonical AhR target gene *PAI-1* (plasminogen activator inhibitor 1) (Figure S4C).

### Carvones influence the cellular functions of AhR
We analyzed in detail the effects of carvones on individual cellular events throughout the AhR signaling pathway. The AhR agonists TCDD, BaP, and FICZ triggered the translocation of AhR from the cytosol to the nucleus, and carvones did not influence this process. Additionally, carvones alone did not induce AhR nuclear translocation (Fig. 3A; Table S3). Nuclear, ligand-bound AhR forms a heterodimer with ARNT, which in turn binds specific dioxin-response elements in the promoters of target genes. This pathway, involving ARNT, is referred to as canonical AhR signaling. Carvones strongly inhibited the formation of AhR-ARNT heterodimers (Fig. 3B) and the binding of AhR to the *CYP1A1* promoter (Fig. 3C) in cells stimulated with TCDD- and BaP- but not with FICZ. We also observed that structural analogs of carvone (monocyclic monoterpenoids) with AhR antagonist activity

(Figure S2) do not inhibit TCDD- and BaP-inducible nuclear translocation of AhR (Figure S5A), but inhibit formation of AhR-ARNT heterodimer (Figure S5B). A scheme summarizing the effects of carvones on canonical AhR signaling is depicted in Fig. 3D.

### Binding of carvones to AhR
A reporter gene assay revealed that carvones are noncompetitive antagonists of AhR (Fig. 1, Figure S1), implying that they should not competitively displace ligands from binding to AhR. This assumption was corroborated by a competitive radioligand binding assay, which showed that S-carvone did not inhibit the binding of [3]H-TCDD to mouse hepatic Ahr. However, we observed a slight, concentration-independent decrease in [3]H-TCDD binding in the presence of 1000 μM S-carvone (Fig. 4A). Noncompetitive antagonism may occur through the following potential mechanisms. (i) Allosteric hindrance (direct or involving conformational change) of the ligand binding pocket of AhR would prevent proper binding of the ligand and switching on of AhR. This scenario is unlikely because the ligand-dependent nuclear translocation of AhR was not disturbed by carvones (Fig. 3A). For this reason, irreversible competitive antagonism is also unlikely. (ii) An indirect mechanism could occur either at AhR or off-target, affecting protein kinases, ARNT, etc. (*vide infra*). Therefore, we investigated the allosteric binding of carvones to AhR and their effects on AhR-ARNT

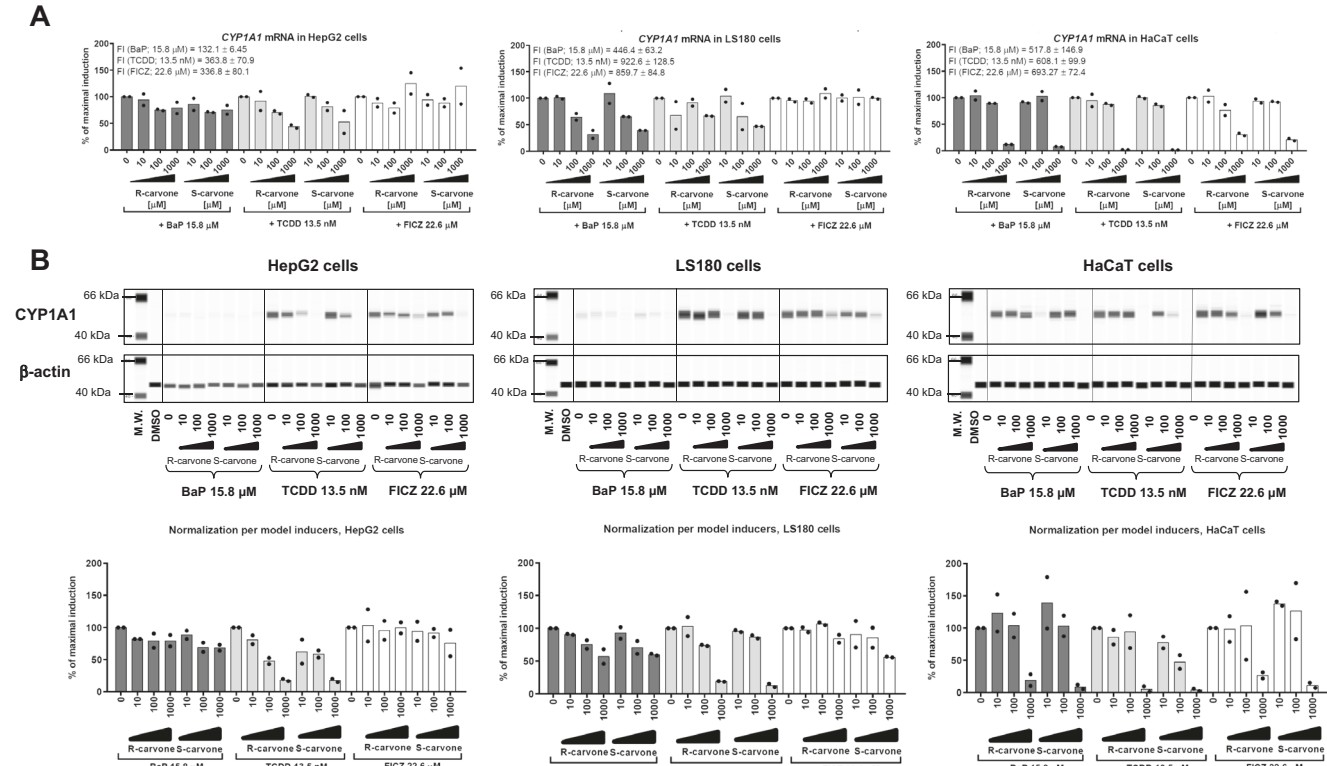

**Fig. 2 | Downregulation of CYP1A1 in human cell lines by carvones.** HepG2, LS180, and HaCaT cells were incubated for 24 h with carvones (0 µM – 1000 µM) in the presence of the AhR agonists TCDD, BaP and FICZ at their $EC_{80}$ concentrations. Incubations and measurements were performed in triplicate (technical replicates). **(A)** RT-PCR analyses of *CYP1A1* mRNA; results expressed relative to the agonist in the absence of carvones (100%). Data are the mean from two independent cell passages (n = 2). The results were normalized using *GAPDH* as a housekeeping gene. The absolute values of *CYP1A1* mRNA fold inductions (F.I.) by model agonists are indicated in the text inserted in the bar graphs from each cell line. **(B)** Quantitative automated Western blot analysis by SallySue of CYP1A1 protein. Representative SallySue records from one cell passage are shown. Bar graphs at the bottom show quantified CYP1A1 protein normalized *per* β-actin; data are expressed relative to agonist in the absence of carvones (100%) and are the mean ± S.D. from two independent cell passages (n = 2). Abbreviations: BaP, benzo[*a*]pyrene; FICZ, 6-formylindolo[3,2-*b*]carbazole; TCDD, 2,3,7,8-tetrachlorodibenzo-*p*-dioxin. Source data are provided as a Source Data file.

heterodimerization. The molecular docking of carvones to various known binding pockets of AhR ligands, such as TCDD, resveratrol, FICZ, BaP, and methylindoles, suggested that carvones may nonspecifically bind to these sites with average docking scores of 47.5 and 42, respectively. However, this binding could be due to their relatively small size and could have no functional effect. Based on the experimental evidence that carvones inhibit the formation of AhR-ARNT (Fig. 3B), we docked these molecules to the heterodimerization interface of AhR and ARNT. This interface spans several interdomain interactions that also form the dioxin-responsive element binding pocket[24]. One such interface region is the α1-α2 helical region of the bHLH domain consisting of residues Leu43, Leu47, and Leu50 from the α1 helix and Tyr76, Leu72, and Leu77 from the α2 helical region. Carvones were docked to the interface site, and the complex of AhR with carvones was simulated for ~250 ns to allow the ligand to dock stably to AhR (Figure S7). Carvones bind favorably at a site formed by residues from the bHLH domain, including close contacts with Tyr76, Pro55, Phe83, Tyr137, Leu72, Pro91, Lys80, Ala79 and Phe136 (Figure S7). More significantly, the binding of carvones to this site shifts the position of both the α1 and α2 helical regions by 10–13 Å and significant unwinding of α1 helix (Figure S7), which can affect the formation of the AhR-ARNT complex. By means of microscale thermophoresis using bacterially coexpressed fragments of human AhR (23-273) and mouse Arnt, we showed that carvones bind AhR but not Arnt (Fig. 4B). Although the binding of carvones to AhR was concentration dependent, the apparent binding constant $K_D$ could not be determined since it lays in the low millimolar range, probably due to artificial conditions using truncated variants of AhR and Arnt. The interaction of carvones with AhR fragment spanning from amino acids

23 to 273, implies that the binding of carvones was localized outside the conventional ligand binding domain but within the bHLH/PAS-A region of AhR. These data fully support the hypothesis that carvones' non-competitive antagonism involves allosteric binding to AhR. In addition, D-limonene (deoxo analog of carvone) did not display AhR antagonism and did not bind human AhR (23-273), which reveals the importance of the oxo moiety in the carvone molecule for interaction with AhR, tentatively through hydrogen bonds (Figure S6A–C). We also observed non-specific displacement of [3H]-TCDD from mouse Ahr by high concentrations of D-limonene (Figure S6D). To identify AhR amino acid residues involved in the interaction with carvones, we employed multiple approaches: (i) Thermal shift analyses using purified and cellular human AhR PAS-A domain (112-272) showed that carvones do not interact with PAS-A region of AhR (Fig. 4C). Since carvones bind AhR (23–273) fragment, as showed by microscale thermophoresis (Fig. 4B), it is likely that amino acid residue critical for carvone interaction with AhR is located in the region (23–111); (ii) Therefore, and taking in account the candidate residues identified by docking, we bacterially expressed and purified His-AhR (23–273) proteins mutated at Tyrosine 76 (Y76A, Y76F). Using microscale thermophoresis, we observed that binding of carvones was completely lost or strongly diminished in Y76A and Y76F mutants, respectively, as compared to wild type AhR (Fig. 4D); (iii) Finally, we used a technique of protein cross-linking with a photo-activated azido-labeled ligand. We synthesized $N_3$-S-carvone (Figure S6E) and confirmed that it still exhibits AhR-antagonist capability against TCDD- and BaP-activated AhR (Figure S6F). By using MALDI, the recombinant AhR was identified in the control samples by 19 assigned tryptic peptides in the *m/z* range of 578.3–5207.5, which covered the

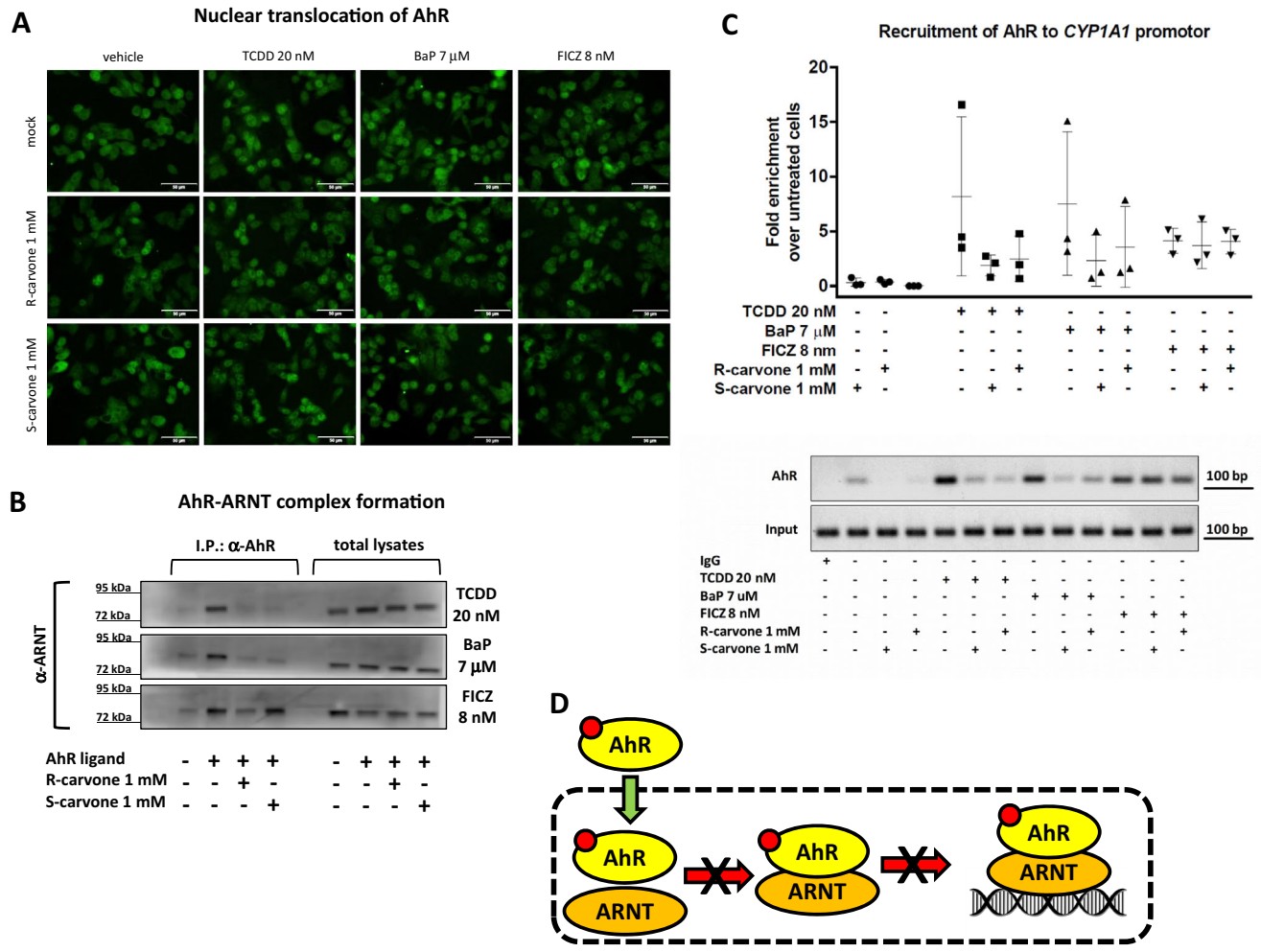

**Fig. 3 | Carvones influence the cellular functions of AhR.** Cells were incubated for 90 min with carvones (1000 μM) in combination with vehicle (0.1% DMSO) or AhR agonists TCDD (20 nM), BaP (7 μM), or FICZ (8 nM). **(A)** The nuclear translocation of AhR is not influenced by carvones. Microscopic specimens from LS180 cells were prepared using Alexa Fluor 488-labeled primary antibody against AhR and DAPI. AhR was visualized and evaluated using a fluorescence microscope. Experiments were performed in two consecutive cell passages (n = 2) with all tested compounds in duplicate. Representative images are shown. Scale bar = 50 μM. **(B)** Carvones inhibit the formation of AhR-ARNT heterodimers. Protein coimmunoprecipitation – formation of AhR-ARNT heterodimers in LS180 cells. Representative immunoblots of immunoprecipitated protein eluates and total cell lysates are shown.

Experiments were performed in three consecutive cell passages (n = 3). **(C)** Carvones inhibit the binding of AhR to the *CYP1A1* promoter. Chromatin immunoprecipitation ChIP – binding of AhR to the *CYP1A1* promoter in HepG2 cells. The bar graph (top) shows enrichment of the *CYP1A1* promoter with AhR compared to vehicle-treated cells. Representative DNA fragments amplified by PCR analyzed on a 2% agarose gel are from the 2nd experiment (bottom). Experiments were performed in three consecutive cell passages (n = 3). Data are shown as mean ± SD. **(D)** Schematic depiction of the cellular effects of carvones binding to AhR. Abbreviations: BaP, benzo[*a*]pyrene; FICZ, 6-formylindolo[3,2-*b*]carbazole; TCDD, 2,3,7,8-tetrachlorodibenzo-*p*-dioxin. Source data are provided as a Source Data file.

regions of 33–107 and 168–250, representing a major part of the recombinant protein and approximately 17% of the full-length native AhR sequence (Swiss-Prot database accession number AHR_HUMAN P35869). Treatment with photoactivated $N_3$-S-carvone yielded a tiny difference in the representation of peptide peaks. Two peptides were absent in the peptide fingerprint of AhR reacted with $N_3$-S-carvone compared to the control (Fig. 4D): *m/z* 1036.6 and 1131.5, with the amino acid sequences 72-LSVSYLRAK-80 and 41-DRLNTELDR-49, respectively (confirmed by MS/MS sequencing). The former peptide has an alternative and preferred cleavage form, LSVSYLR, observable as a high peak at *m/z* 837.5, indicating a low affinity-labeling yield. A low-intensity peak at *m/z* 1199.6 corresponded to a mass difference of 163 Da, reflecting to the binding of the $N_3$-S-carvone-derived nitrene moiety at LSVSYLRAK. However, due to the low intensity, we could not obtain direct confirmation by MS/MS sequencing. The photoactivation process itself performed well because $N_3$-S-carvone, upon MALDI with the UV laser (measured in the presence of cetrimonium bromide in the matrix solution according to Guo et al.[25]), exhibited a peak at *m/z* 164,

indicating the loss of molecular nitrogen from the azido group. Altogether, considering data from molecular docking, microscale thermophoresis (S-carvone, D-limonene) and AhR covalent functionalization with photoactivated $N_3$-S-carvone, Tyr76 is highly likely to play a key role in the allosteric binding of S-carvone to AhR.

## S-carvone does not inhibit a random panel of protein kinases, including PKC

Blocking protein kinase C (PKC) activity was reported to inhibit the transcription of *CYP1A1* but to exert no effect on the nuclear translocation of AhR[26], which was also the case here with carvones. Therefore, we tested whether carvones inhibit PKC catalytic activity. We did not observe any decline in PKC activity measured in lysates from HepG2 cells incubated with carvones at concentrations up to 1000 μM, which rules out the involvement of PKC inhibition in the effects of carvones on AhR (Fig. 5A). We also evaluated the interaction between 100 μM S-carvone and 468 human protein kinases, employing KINOMEscan™ (scanMAX assay), a proprietary active site-directed competition

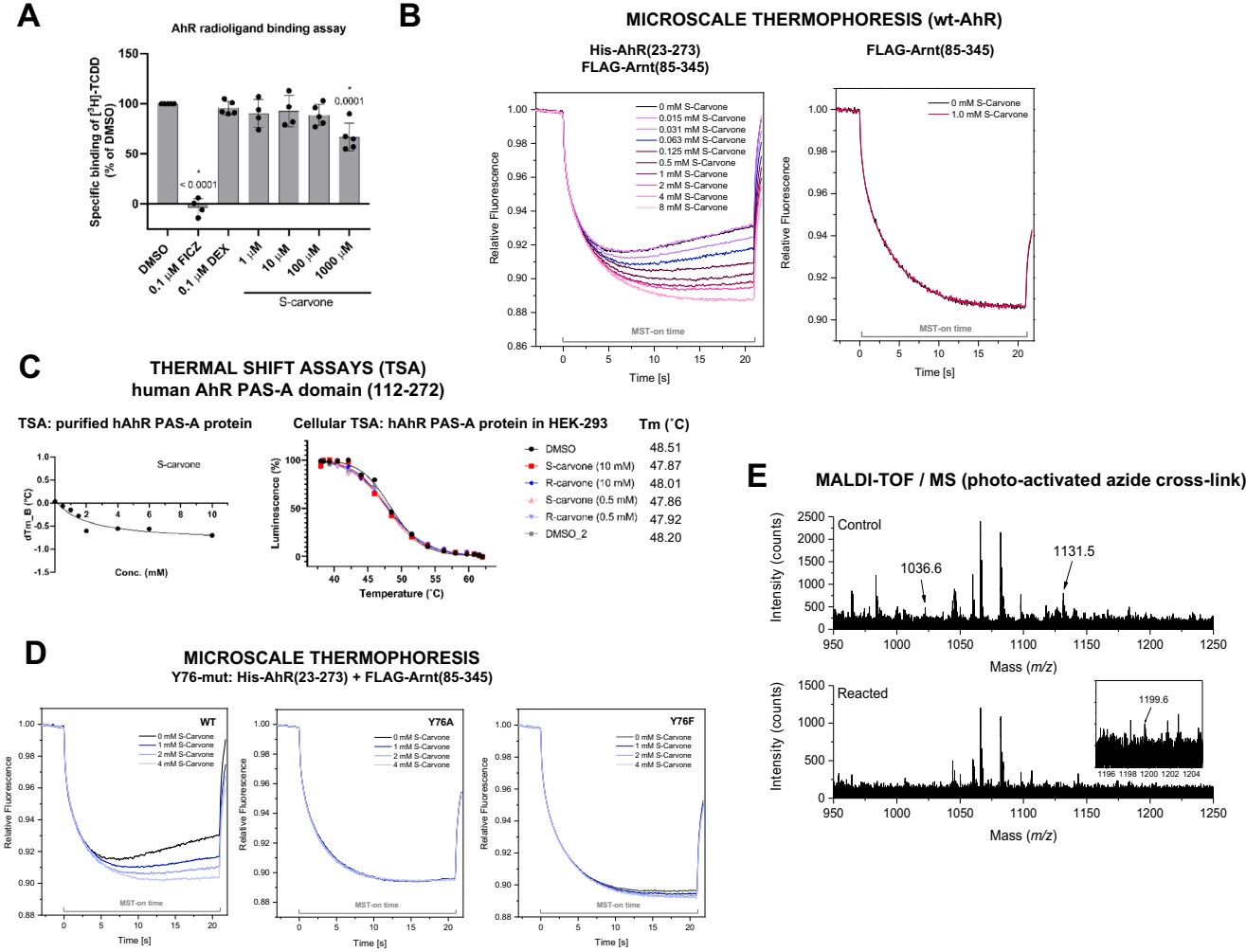

**Fig. 4 | Binding of S-carvone to AhR. A** Competitive radioligand binding assay: Cytosolic protein (2 mg/mL) from Hepa1c1c7 cells was incubated with S-carvone (1 μM, 10 μM, 100 μM, 1000 μM), FICZ (10 nM), DEX (100 nM; negative control) or DMSO (0.1% V/V; corresponding to *specific binding of [³H]-TCDD = 100%*) in the presence of 2 nM [³H]-TCDD. Specific binding of [³H]-TCDD was determined to be the difference between total and nonspecific (200 nM; 2,3,7,8-tetrachlorodibenzofuran) reactions. The significance was calculated using 1way ANOVA multiple comparison test, and *p*-values are indicated in the graph. Four independent experiments (*n* = 4) were performed, and the incubations and measurements were performed in triplicate in each experiment (technical replicates). The error bars represent the mean ± SD. **B** Microscale thermophoresis; *Left panel:* coexpressed His-AhR(23-273) + FLAG-Arnt(85-345) incubated with S-carvone (0.25 mM, 0.5 mM, 1 mM, 2 mM, 4 mM); *right panel:*. FLAG-Arnt(85–345) incubated with vehicle or 1 mM S-carvone. **C** *Left panel:* Thermal shift assay with purified hAhR PAS-A (112-272) protein incubated with S-carvone (0.5 mM, 1 mM, 1.5 mM, 2 mM, 4 mM, 6 mM, 10 mM); *right panel*: Cellular thermal shift analyses in Hek293 cell transfected with hAhR PAS-A (112-272) protein; incubations with S/R-carvones (0.5 mM and 10 mM). **D** Microscale thermophoresis: coexpressed (WT; mutY76A; mutY76F)-His-AhR(23-273) + FLAG-Arnt(85-345) incubated with S-carvone (0.25 mM, 0.5 mM, 1 mM, 2 mM, 4 mM). **E** MALDI-TOF MS of peptides from AhR tryptic digests: The top panel shows a detail of the control digest spectrum. The two labeled peptides (*m/z* 1036.6 and 1131.5) were absent after the reaction of AhR with photoactivated N₃-S-carvone (bottom panel). The inset shows a close-up view of the reacted AhR digest spectrum to demonstrate the presence of a putatively modified peptide at *m/z* 1199.6, which might be related to that at *m/z* 1036.6 in the control. *FICZ* 6-formylindolo[3,2-*b*]carbazole, *DEX* dexamathasone, *DMSO* dimethylsulfoxide. Source data are provided as a Source Data file.

binding assay[27]. The minimal inhibitory threshold used by the screening platform KINOMEscan™ is 35% of the control kinase activity. Among 468 kinases tested, 467 were above the 35% cutoff. Tyrosine kinase 2 TK2 (JH2 domain pseudokinase) activity was inhibited to 27% of the control activity, but this kinase is not relevant to the regulation of transcription activity (Fig. 5B; Figure S9). Overall, we excluded the possibility that the effects of carvones on AhR signaling are caused indirectly by the inhibition of the human kinome, particularly PKC.

## Carvones do not inhibit ARNT transcriptional activity
ARNT is involved in other cellular pathways in addition to AhR, such as hypoxia signaling, which is transcriptionally mediated by the ARNT heterodimer with hypoxia-inducible factor 1α (HIF1α). Therefore, we investigated the effects of carvones on the hypoxia-mimic inducible, ARNT-dependent expression of vascular endothelial

growth factor (*VEGF*) in HaCaT keratinocytes incubated with deferoxamine. The levels of *VEGF* mRNA were increased 5-fold by deferoxamine, and carvones at concentrations up to 1000 μM did not influence this induction. Consistently, the hypoxia-mimic decrease in *CYP1A1* mRNA was not affected by carvones (Fig. 5C). These data imply that carvones do not inhibit ARNT transcriptional activity and that disruption of AhR-ARNT complex formation is not due to the interaction of carvones with ARNT. These observations corroborate the finding that carvones do not bind truncated recombinant Arnt (Fig. 4B).

## Carvones do not interact with AhR-related off-targets
Using radioligand binding assays, we showed that S-carvone (up to 1000 μM) does not bind to human nuclear and steroid receptors that transcriptionally cross-talk with AhR, including androgen receptor

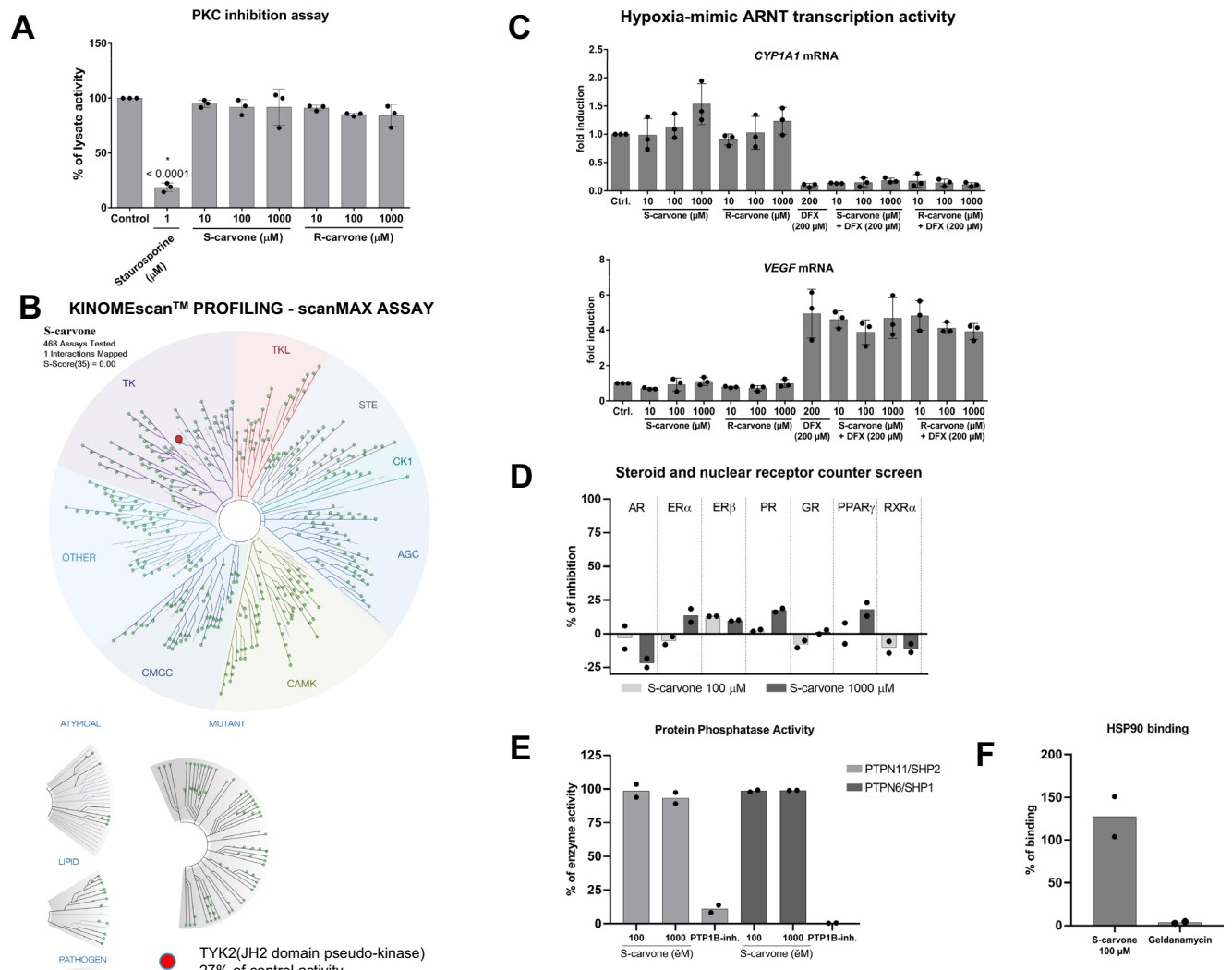

**Fig. 5 | Evaluation of off-target effects of carvones. A** PKC inhibition assay: The catalytic activity of PKC was measured in lysates from HepG2 cells incubated with vehicle (DMSO, 0.1% V/V), staurosporine (1 μM), and carvones (10 μM; 100 μM; 1000 μM). Data are the mean ± SD from three independent experiments (n = 3). The significance was calculated using 2-way ANOVA multiple comparison test, and p-values are indicated in the graph. Incubations and measurements were performed in uniplicate (technical replicates). **B** KINOMEscan™ profiling: The interaction between 100 μM S-carvone and 468 human protein kinases, employing KINOMEscan™ (scanMAX assay), a proprietary active site-directed competition binding assay. A low-resolution interaction map is shown. **C** Hypoxia-mimic *VEGF* induction: HaCaT cells were incubated for 24 h with carvones (10 μM; 100 μM; 1000 μM) in combination with vehicle (0.1% DMSO) or deferoxamine (DFX; 200 μM). The expression of *VEGF* and *CYP1A1* mRNAs was measured using RT-PCR. Incubations and measurements were performed in duplicate (technical replicates). Data are the mean ± SD from three independent cell passages (n = 3) and are expressed as fold induction over the vehicle-treated cells. The results were normalized using *GAPDH* as a housekeeping gene. **D** Counterscreen radioligand binding assay with human AR, ERα, ERβ, PR, GR, RXRα, and PPARγ receptors. The bar graph shows the percentage of displacement of the radioligand by S-carvone (100 μM; 1000 μM). The data are mean from two independent incubations (n = 2). A value >50% is considered to indicate interaction. **E** Protein phosphatase inhibition assay: The catalytic activity of PTPN11/SHP2 and PTPN6/SHP1 was measured with recombinant enzymes incubated with vehicle (DMSO, 0.1% V/V), PTP1B inhibitor (33.3 μM), and S-carvone (100 μM; 1000 μM). Data are the mean from two independent experiments (n = 2). Incubations and measurements were performed in duplicate (technical replicates). **F** Competitive fluorescence binding assay for HSP90: The bar graph shows the percentage displacement of fluorescently labeled geldanamycin by S-carvone (100 μM) and nonlabeled geldanamycin (0.12 μM). Data are the mean from two independent experiments (n = 2). Incubations and measurements were performed in duplicate (technical replicates). *AR* androgen receptor, *ARNT* AhR nuclear translocator, *DFX* deferoxamine, *ERα/β* estrogen receptor alpha/beta, *GR* glucocorticoid receptor, *HSP90* heat shock protein 90 kDa, *PKC* protein kinase C, *PPARγ* peroxisome proliferator-activated receptor gamma, *PR* progesterone receptor, *PTPN6(11)/SHP1(2)* tyrosine-protein phosphatases non-receptor type 6(11), *RXRα* retinoid X receptor alpha, *VEGF* vascular endothelial growth factor. Source data are provided as a Source Data file.

(AR), retinoid X receptor alpha (RXRα), glucocorticoid receptor (GR), progesterone receptor (PR), estrogen receptors alpha/beta (ERα/β) and peroxisome proliferator-activated receptor gamma (PPARγ) (Fig. 5D). S-carvone did not inhibit the catalytic activity of the non-receptor-type tyrosine-protein phosphatases PTPN11/SHP2 and PTPN6/SHP1, which are critical in regulating the AhR stress response (Fig. 5E). Finally, S-carvone did not displace geldanamycin from binding to heat shock protein 90 kDa, which is a cytosolic binding partner of AhR (Fig. 5F).

## Carvones antagonize Ahr in mouse skin

We carried out a study on live mouse skin to determine the antagonistic effects of carvones on ligand-activated Ahr (Fig. 6A). The topical application of BaP to mouse outer ears (auricles) induced Cyp1a1 mRNA expression, and this induction was inhibited (62% inhibition) in mouse ears preincubated with S-carvone (Fig. 6B). The involvement of Ahr in skin inflammation and the modulation of inflammatory mediators by Ahr ligands has been reported[28]. Irradiation of mouse ears with ultraviolet (UV) light caused the induction of

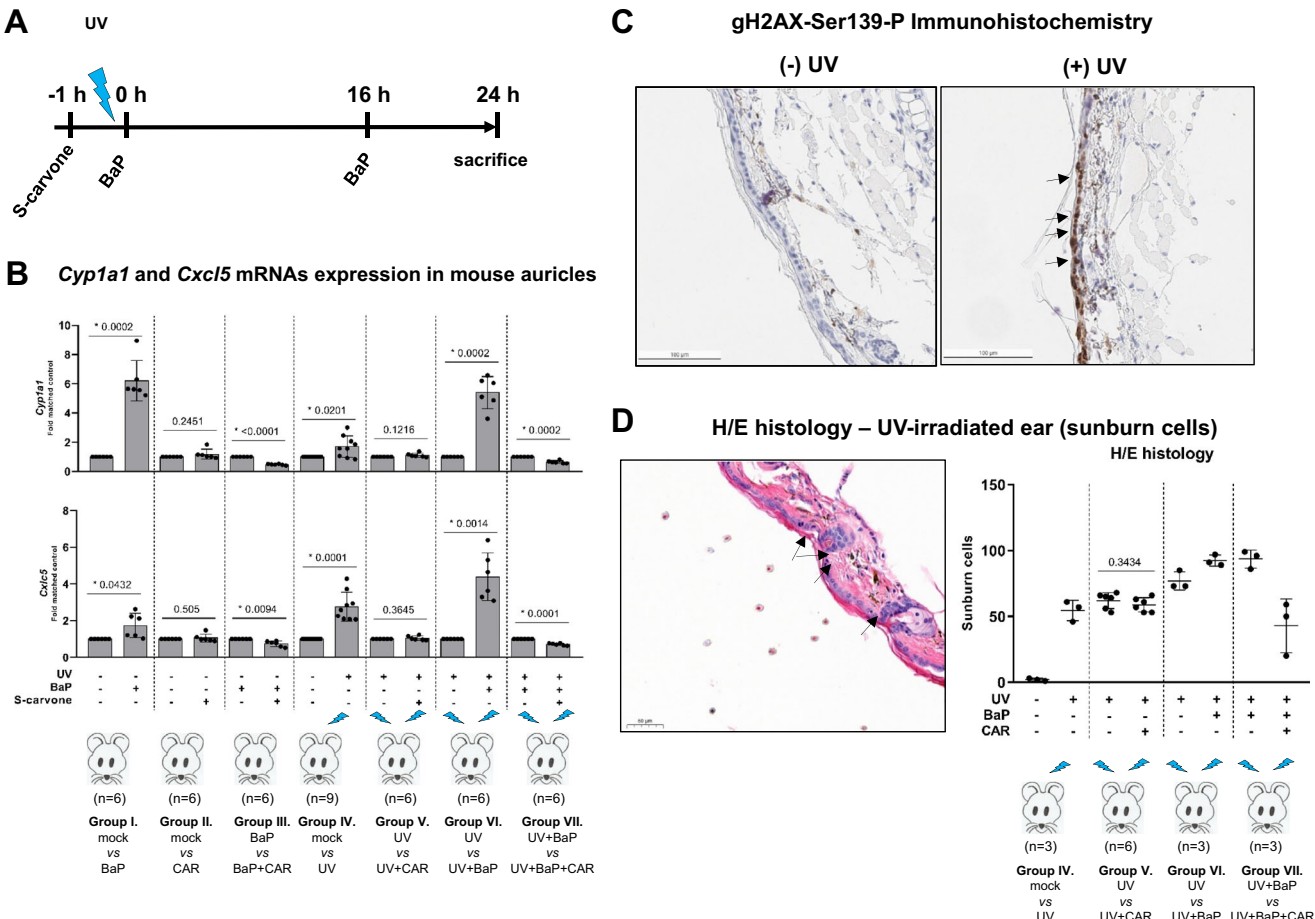

**Fig. 6 | Carvones antagonize Ahr in vivo.** Tested compounds and UV irradiation were applied to C57BL/6 mouse auricles as described in the Methods section. The left and right ears were exposed to different treatments, thereby providing internal individual controls for comparative treatments, and each animal represents a biological repeat. **A** Scheme of the treatment. **B** RT-PCR analyses of *Cyp1a1* and *Cxcl5* mRNAs were performed in quadruplicate. The results are expressed as a ratio between the left and right ears of an individual mouse. Data are the mean ± SD from at least six mice *per* treatment group. The results were normalized using *Rplp0* as a housekeeping gene. The data normality was tested by Shapiro-Wilk test. The significance was calculated using one-sample two-tailed t-test, and *p*-values are indicated in the graph. **C** Immunohistochemistry: Representative control and UV-exposed mouse ear tissues stained with an antibody against γH2AX-Ser-P are shown. The analyses were performed in ear tissue from six mice (*n* = 6). Scale bar = 100 μM. **D** Histology: Representative UV-exposed mouse ear tissue stained with hematoxylin & eosin is shown. The scatter plot shows the count of UV-burned cells in the left and right ears of an individual mouse. Data are the mean ± SD from three mice *per* treatment group. The significance was calculated using paired two-tailed t-test, and *p*-value is indicated in the group UV *vs* UV+carvone group (*n* = 6). Scale bar = 50 μM. BaP benzo[*a*]pyrene, CAR S-carvone, *CYP1A1* cytochrome P450 1A1, *CXCL5* C-X-C motif chemokine 5, *H/E* hematoxylin/eosin stain. Source data are provided as a Source Data file.

the proinflammatory chemokine *Cxcl5*, and postirradiation application of BaP further increased the levels of UV-inducible *Cxcl5*. Carvones reversed (38% inhibition) the induction of *Cxcl5* levels by BaP in UV-irradiated mouse ears (Fig. 6B), which is of potential clinical importance because *Cxcl5* was identified as a target gene associated with UV irradiation-induced inflammatory pain in sunburn subjects[29]. Immunohistochemical staining with γH2AX-Ser139-P antibody confirmed widespread DNA damage[30] in UV-irradiated mouse ears (Fig. 6C). The aggravating effect of BaP on UV-induced tissue damage was attenuated by carvones, as revealed by the sunburn cell count[31] of hematoxylin/eosin-stained histology samples (Fig. 6D). Of note, S-carvone itself did not influence the number of UV-burned cells, regardless of its application prior (Fig. 6D) or post (Figure S8) UV-exposure. These data disprove potential shielding effect of S-carvone against UV-irradiation.

## Discussion

Therapeutic targeting of AhR has long been neglected, mainly due to the negative impression of AhR being a receptor mediating TCDD toxicity. With increasing knowledge of the physiological and pathophysiological roles of AhR, attempts to target AhR have emerged,

including the therapy of cancer, inflammatory bowel disease, and atopic dermatitis. The following strategies are employed: (i) repositioning of clinically used AhR-active drugs (e.g., tranilast, flutamide, omeprazole); (ii) chemoprevention with dietary AhR-active compounds (e.g., indole-3-carbinol, diindolylmethane); and (iii) the application of novel AhR ligands identified by screening chemical libraries (e.g., CH223191)[8] or by rational design (e.g., PY109)[32].

The interactions between small-molecule compounds and AhR signaling may occur directly (ligand-dependent) or indirectly through off-targets such as PKC[26], protein tyrosine kinases[33], or cAMP[34]. To date, all reported AhR ligands, both agonists and antagonists, are orthosteric ligands, i.e., ligands that bind to a conventional discrete site on the AhR protein, referred to as the ligand binding pocket. Recently, three structurally distinct groups of AhR orthosteric ligands were defined according to the mode of their interactions with residues within the AhR ligand binding site[35]. Functionally, the effects of AhR ligands comprise full agonists, partial agonists, and competitive antagonists. This report describes small-molecule compounds acting as allosteric antagonists of human AhR, which may potentially be of clinical importance. Allosteric approaches to inhibiting receptors

circumvent any dependence on ligand binding or the nature of the ligand involved[36].

Applying a series of complementary mechanistic experiments, we demonstrated that carvones are noncompetitive antagonists of human AhR, acting through allosteric binding in the region of the AhR involved in heterodimerization with ARNT, thereby preventing the formation of functional AhR-ARNT heterodimers. In brief, detailed analyses of the AhR transcriptional response in reporter gene assays revealed a noncompetitive mechanism of carvone antagonism. This is consistent with the finding that carvones did not displace [3]H-TCDD from binding AhR and did not inhibit the ligand-elicited nuclear translocation of AhR. On the other hand, carvones inhibited the formation of AhR-ARNT heterodimers, and all downstream events involving the binding of AhR to DNA and the expression of AhR target genes. We excluded the interactions of carvones with potential off-targets, including ARNT, PKC, a panel of 468 kinases, steroid receptors (AR, ERα, ERβ, PR, GR), nuclear receptors (RXRα, PPARγ), protein phosphatases (PTPN11/SHP2, PTPN6/SHP1) and heat shock protein 90 kDa (HSP90). As a proof of concept, we demonstrated the AhR antagonist effects of carvones in vivo. First, carvones inhibited the BaP-inducible xenobiotic-metabolizing *Cyp1a1* in mouse skin. Second, carvones reversed the BaP-mediated potentiation of the UV-inducible chemokine *Cxcl5*, which was previously found to be accountable for UVB irradiation-induced inflammatory pain[29]. In our exploratory proof-of-concept and hypothesis generation experiment, carvones also attenuated the harmful effects of BaP, as revealed by the sunburn cell count. In UVB-irradiated skin, AhR signaling pathways are activated by FICZ and related tryptophan photoproducts and facilitate cutaneous inflammatory responses and skin carcinogenesis[37]. We have earlier shown that a topical application of the competitive antagonist BDDI inhibits AHR dependent signaling in UVB-irradiated human skin[38]. Our current data indicate that the topical application of carvones to sunburned skin might reduce Cxcl5-related inflammatory symptoms and pain.

Differential roles of the AhR and ARNT residues in molecular events preceding and following heterodimerization of AhR with ARNT were determined by Corrada et al.[39,40]. The crystal structure of the mouse Ahr PAS-A domain revealed that mouse Ahr residues Ala119 and Leu120 are critically crucial for hydrophobic interactions at the Ahr-Arnt interface and dimerization[41]. Seok et al. determined the crystal structure of a mouse Ahr-Arnt heterodimer in complex with DRE, showing that Arnt curls around Ahr into a highly intertwined asymmetric architecture, with extensive heterodimerization interfaces and interdomain interactions within Ahr. They proposed the phenomenon of ligand-selective structural hierarchy for complex scenarios of Ahr activation[24]. Mutations in mouse Ahr residues Leu42 and Leu120 (homologous to human Leu43 and Leu122) led to decreased binding of Ahr-Arnt to DRE[24], which corroborates the findings of Wu et al.[41]. The mutation of Leu49 in mouse Ahr maintained the nuclear translocation of Ahr but inhibited its transcriptional activity[24], which was mimicked by the binding of carvones to AhR. According to our docking data, carvones interact with residues in the bHLH domain, and the proposed docking models show that the binding of carvones directly affects the dimerization interface with a helical shift of 10–13 Å and significant unwinding of the α1 helix (Figure S7). This assumption was experimentally confirmed, and direct binding of carvones at the AhR fragment spanning from amino acids 23 to 273 was demonstrated using microscale thermophoresis. The significance of the oxo moiety in the carvone molecule for interaction with AhR, tentatively through hydrogen bonds, was demonstrated because deoxy-carvone (D-limonene) did not bind the AhR fragment and did not antagonize AhR. By using the Achilles Blind Docking Server (http://bio-hpc.eu/software/blind-docking-server/[42]), we calculated that the target residues for the photoactivable derivative N₃-S-carvone include Leu47, Phe56, Leu72

and Tyr76. Indeed, we demonstrated that the peptides absent from the AhR digest after the reaction with N₃-S-carvone contained the predicted sites (Fig. 4E). Microscale thermophoresis using Y76A and Y76F mutants of AhR(23-273) revealed the importance of tyrosine Y76 for interaction with carvones. The biological effects of carvones against AhR were attained at concentrations spanning from 100 μM to 1000 μM, which might appear high; however, the available data suggest that these concentrations are relevant. The topical application of 300 mg of R-carvone or S-carvone, which are used as skin permeabilizers in transdermal patches, resulted in maximal plasma concentrations of ~0.6 μM and ~0.2 μM, respectively[20]. Moreover, the local concentrations of carvones in the skin, after topical application must be orders of magnitude higher than the plasma levels. The blood levels of carvone in volunteers who received 100 mg of caraway oil (~54.5 mg carvone) in coated capsules reached approx. 0.1 μM[43]. However, local concentrations of carvones in the intestines (intestinal first-pass) and liver (hepatic first-pass) must be much higher than those reached in the plasma. The concentration of carvones in foods is approximately 150 μM, implying the exposure of enterocytes to such concentrations when consuming food containing EOs of caraway, spearmint, or dill[44]. The European Food Safety Authority EFSA defined the acceptable daily intake of S-carvone as 0.6 mg/kg of body weight. In addition, a recent estimate based on the recommended dose and the published fecal excreted fraction of 200 marketed drugs reported a median expected colon concentration of 80 μM for drugs having a median serum concentration of 0.6 μM, implying globally >100-fold higher drug concentrations in the gut compared to blood[45]. Collectively, the potential clinical or preventive use of carvones as AhR antagonists is indicated by their local (not systemic) effects on the skin (topical application) or in the intestines (*per os* intake).

In summary, we report dietary monocyclic monoterpenoid carvones, as noncompetitive antagonists of AhR that act through allosteric binding to AhR, thereby blocking heterodimerization with ARNT and constraining the transcriptional functions of AhR-ARNT. While hundreds of orthosteric AhR ligands, including antagonists, have been described, here we report the allosteric antagonism of AhR by small-molecule compounds, which might be of both clinical and fundamental mechanistic importance.

## Methods

The research complied with all relevant ethical regulations; (i) Human hepatocytes cultures−the tissue acquisition protocol issued by the "Ethical Committee of the Faculty Hospital Olomouc, Czech Republic"; (ii) Animal experiments−approved by the Institutional Animal Care and Use Committee of the Albert Einstein College of Medicine (New York, NY, USA; Protocol #00001405).

### Chemicals and materials

S-carvone (sc-239480, purity 99.4%, Lot L0613), R-carvone (sc-293985, purity 99.7%, Lot H1015), indole (sc-257606, purity 98.4%, Lot J0918), 3-methylindole (sc-256535, purity 99.9%, Lot E0418), indirubin (sc-201531A, purity 96.5%, Lot B1513), and D-limonene (sc-205283, Lot F1314) were purchased from Santa Cruz Biotechnology. BaP (B1760, Lot SLBS0038V, purity 99%), FICZ (SML1489, Lot 0000026018, purity 99.5%), staurosporine (S4400, purity 98%), deferoxamine mesylate (DFX; D9533, purity 92.5%), cuminol (W293318, purity 99.7%, Lot MKCF1644), p-cymene (C121452, purity 99.4%, Lot MKCG5957), cuminaldehyde (#135178, purity 99.6%, Lot MKCC7110), (+)-dihydrocarvone (#218286, purity 99.9%, Lot MKBN2588v), R-pulegone (#376388, purity 99.7%, Lot MKCF5536), S-pulegone (#328847, purity 98.0%, Lot 1433568), thymol (T0501, purity 99.5%, Lot SLBZ9699), carvacrol (W224511, purity 99.9%, Lot MKCG2266), (+)-menthone (#63675, purity 98.6%, Lot BCCC1798), (-)-menthone (W266701, purity 98.7%, Lot STBH6314), piperitone (#79899, purity 98.3%, Lot BCBZ9676), and dexamethasone (DEX; D4902, Lot 112K12845, purity 98%) were obtained

from Sigma-Aldrich (Prague, Czech Republic). TCDD (RPE-029) was purchased from Ultra Scientific, and 2,3,7,8-tetrachlorodibenzofuran (TCDF; Amb17620425, Lot 51207-31-9) was obtained from Ambinter. (+)-isomenthone (M199545, purity 96.7%, Lot 1-NKM-85-3), and (-)-isomenthone (M199560, purity ~80%, Lot 1-NKM-97-3) were obtained from Toronto Research Centre Inc. (Toronto, Canada). Piperitenone (#491-09-8, purity 95.0%, Lot BS17ZJ05212) was acquired from BOC Sciences (Shirley, NY, USA). Radiolabeled [³H]-TCDD (ART 1642, Lot 181018, purity 98.6%) was purchased from American Radiolabeled Chemicals. Bio-Gel® HTP hydroxyapatite (1300420, Lot 64079675) was obtained from Bio-Rad Laboratories.

### Cell lines and hepatocytes
The human hepatoma cell line HepG2 (ECACC No. 85011430), the intestinal human colon adenocarcinoma cell line LS180 (ECACC No. 87021202), the human immortalized keratinocyte line HaCaT (kindly donated by P. Boukamp, IUF Düsseldorf, Germany), and the mouse hepatoma cell line Hepa1c1 (ECACC No. 95090613) were cultured as recommended by the supplier. The primary human hepatocytes LH75 (female, 78 years, Caucasian) were prepared at the Faculty of Medicine, Palacky University Olomouc. The tissue acquisition protocol complied with the regulation issued by the "Ethical Committee of the Faculty Hospital Olomouc, Czech Republic" and Transplantation law #285/2002 Coll (which implies presumed consent from donor). The primary human hepatocytes Hep200571 (male, 77 years, unknown ethnicity) and Hep220993 (female, 76 years, Caucasian) were purchased from Biopredic International (Rennes, France). Mycoplasma Detection Kit-Digital Test v2.0 Cat. No. B39132 (Biotool) was used to survey mycoplasma infection. The study involving human tissue samples was conducted in accordance with the declaration of Helsinki.

### Reporter gene assays
The stably transfected gene reporter cell line AZ-AHR, derived from the human hepatoma cell line HepG2, expressing endogenous AhR and transfected with a construct containing several AhR binding sites upstream of a luciferase reporter gene, was used to evaluate the transcriptional activity of AhR[21]. Cells were seeded in 96-well culture plates, and after 16 h of stabilization, they were incubated for 4 h and 24 h with the tested compounds or their combinations. Thereafter, the cells were lysed, and luciferase activity was measured on a Tecan Infinite M200 Pro plate reader (Schoeller Instruments, Czech Republic). The half-maximal inhibitory concentrations ($IC_{50}$), half-maximal effective concentrations ($EC_{50}$), and concentrations of $EC_{80}$ were calculated using GraphPad Prism 8 software (GraphPad Software, San Diego, U.S.A.). Experiments were performed in three independent cell passages. Incubations and measurements were performed in quadruplicate (i.e., four technical replicates).

### Inhibition of luciferase catalytic activity
Stably transfected AZ-AHR cells and COS-7 cells transiently transfected with DRE-luc reporter plasmid were incubated for 24 h with 20 nM TCDD. Cells were lysed and the lysate containing luciferase was incubated for 30 min with carvones (1000 µM) or vehicle (control). Luciferase activity was measured on a 446 Tecan Infinite M200 Pro plate reader (Schoeller Instruments, Czech Republic).

### Cell viability−MTT test
Cells were incubated for 24 h with carvones, vehicle (DMSO; 0.1% v/v) and Triton X-100 (1%, v/v), using multi-well culture plates of 96 wells. MTT test was performed and absorbance was measured spectrophotometrically at 540 nm on Infinite M200 (Schoeller Instruments, Prague, Czech Republic). The data were expressed as the percentage of cell viability, where 100% and 0% represent the treatments with the vehicle and Triton X-100, respectively.

### Quantitative real-time polymerase chain reaction qRT-PCR
Total RNA from cultured cells was isolated using TRI Reagent® (Sigma-Aldrich). cDNA was synthesized from 1 µg of total RNA using M-MuLV Reverse Transcriptase and Random Primers 6 (both New England Biolabs) at 42 °C for 60 min and diluted at a 1:4 ratio with PCR grade water. qRT-PCR was carried out on a Light Cycler® 480 Instrument II (Roche). Data were processed by the delta-delta $C_t$ method and normalized to *GAPDH* as a housekeeping gene. In animal experiments, cDNA was synthesized from 2 µg of total RNA using a High Capacity cDNA Reverse Transcription Kit (Thermo Fisher Scientific, #4368814). qRT−PCR was performed using PowerUp SYBR Green Master Mix (Thermo Fisher, #A25742) on a ViiA Y Real-Time PCR System (Thermo Fisher Scientific, USA). The levels of individual mRNAs were determined using the probes and primers listed in Table S1.

### Simple Western blotting by Sally Sue™
The total protein extract was isolated by using ice-cold lysis buffer (150 mM NaCl; 10 mM Tris pH 7.2; 1% (v/v) Triton X-100; 0.1% (w/v) SDS; 1% (v/v) sodium deoxycholate; 5 mM EDTA; anti-protease cocktail; anti-phosphatase cocktail), and the protein concentration was determined using Bradford reagent. CYP1A1 and β-actin proteins were detected by the Sally Sue™ Simple Western System (ProteinSimple™) using Compass Software version 2.6.5.0 (ProteinSimple™). Immunodetection was performed using primary antibodies against CYP1A1 (mouse monoclonal, sc-393979, A-9, dilution 1:100, Santa Cruz Biotechnology) and β-actin (mouse monoclonal, 3700 S, dilution 1:100, Cell Signaling Technology). Detection was performed by horseradish-conjugated secondary antibody followed by a reaction with a chemiluminescent substrate. Full scan blots are available at https://doi.org/10.5281/zenodo.7764002.

### 7-Ethoxyresorufin-*O*-deethylase activity (EROD)
AZ-AhR cells plated in 96-well culture dishes were incubated for 24 h with vehicle (DMSO; 0.1% v/v), TCDD (13.5 nM) and/or S-carvone (1 mM) + TCDD (13.5 nM). After washing with PBS, medium containing 7-ethoxyresorufin (8 µM) and dicoumarol (10 µM) was applied to the cells. Culture plates were incubated at 37 °C for 30 min. After that, an aliquot of 75 µl of the medium was mixed with 125 µl of methanol, and fluorescence was measured in a 96-well plate with 530 nm excitation and 590 nm emission filters using a Tecan Infinite M200 Pro plate reader (Schoeller Instruments, Czech Republic).

### Radioligand binding assays
**Aryl hydrocarbon receptor**. Cytosol from murine hepatoma Hepa1c1c7 cells was isolated as described[46]. Cytosolic protein (2 mg/mL) was incubated for 2 h at room temperature in the presence of 2 nM [³H]-TCDD with S-carvone (1 µM, 10 µM, 100 µM, 1000 µM), D-limonene (10 µM, 100 µM, 1000 µM), FICZ (10 nM; positive control), DEX (100 nM; negative control) or vehicle (DMSO; 0.1% V/V; corresponds to *specific binding of [³H]-TCDD = 100%*). Ligand binding to the cytosolic proteins was determined by the hydroxyapatite binding protocol and scintillation counting. The specific binding of [³H]-TCDD was determined as the difference between total and nonspecific (TCDF; 200 nM) reactions. Five independent experiments were performed, and the incubations and measurements were performed in triplicate (three technical replicates) in each experiment.

A radioligand binding assay counterscreen (with 100 µM and 1000 µM S-carvone) was carried out in a series of human recombinant steroid and nuclear receptors at Eurofins Panlabs Discovery Services Taiwan (New Taipei City, Taiwan) and Eurofins Cerep SA (Poitiers, France):

**Glucocorticoid receptor (GR; NR3C1)**. Endogenous receptor from IM-9 cells. The model ligand was 1.5 nM [³H]-dexamethasone, and the nonspecific competitor was 10 µM triamcinolone. The incubation time was 6 h at 4 °C.

**Androgen receptor (AR; NR3C4).** Endogenous receptor from LNCaP cells. The model ligand was 1 nM [³H]-methyltrienolone, and the non-specific competitor was 1 μM testosterone. The incubation time was 24 h at 4 °C.

**Progesterone receptor (PR; NR3C3).** Endogenous receptor from T47D cells. The model ligand was 0.5 nM [³H]-progesterone, and the nonspecific competitor was 1 μM promegestone. The incubation time was 20 h at 4 °C.

**Estrogen receptor alpha (ERα; NR3A1).** Recombinant receptor expressed in sf9 cells. The model ligand was 0.5 nM [³H]-estradiol, and the nonspecific competitor was 1 μM diethylstilbestrol. The incubation time was 2 h at room temperature.

**Estrogen receptor beta (ERβ; NR3A2).** Recombinant receptor expressed in sf9 cells. The model ligand was 0.5 nM [³H]-estradiol, and the nonspecific competitor was 1 μM diethylstilbestrol. The incubation time was 2 h at 25 °C.

**Peroxisome proliferator-activated receptor gamma (PPARγ; NR1C3).** Recombinant receptor expressed in *E. coli*. The model ligand was 5 nM [³H]-rosiglitazone, and nonspecific competitor was 10 μM rosiglitazone. The incubation time was 2 h at 4 °C.

**Retinoid X receptor alpha (RXRα; NR2B1).** Recombinant receptor expressed in sf9 cells. The model ligand was 5 nM [³H]-9-cis-retinoic acid, and nonspecific competitor was 3 μM 9-cis-retinoic acid. The incubation time was 1 h at 4 °C.

### Intracellular distribution of AhR

An immunofluorescence assay was performed as recently described[47]. Briefly, LS180 cells were seeded on chamber slides (ibidi GmbH, Germany) and cultured for two days. Then, the cells were treated for 90 min with tested compounds in combination with vehicle (0.1% DMSO) or the AhR agonists TCDD (20 nM), BaP (7 μM), and FICZ (8 nM). After treatment, washing, fixation, permeabilization, and blocking, the cells were incubated with Alexa Fluor 488-labeled primary antibody against AhR (sc-133088, Santa Cruz Biotechnology, U.S.A.) diluted 1:500 in 0.5% bovine serum albumin at 4 °C overnight. The next day, nuclei were stained with 4′,6-diamino-2-phenylindole (DAPI), and cells were mounted in VectaShield® Antifade Mounting Medium (Vector Laboratories Inc., USA). AhR translocation into the nucleus was visualized and evaluated using an IX73 fluorescence microscope (Olympus, Japan). The whole staining protocol was performed in two independent experiments in technical duplicates (with all tested compounds). AhR translocation was evaluated visually depending on the distinct signal intensity of the AhR antibody in the nucleus and cytosol. For percentage calculation, approximately one hundred cells from at least four randomly selected fields of view in each replicate were used.

### Protein immunoprecipitation assay

The effects of carvones on the ligand-dependent heterodimerization of AhR with ARNT were studied in cell lysates from LS180 cells incubated with tested compounds in combination with vehicle (0.1% DMSO) or the AhR agonists TCDD (20 nM), BaP (7 μM) and FICZ (8 nM) for 90 min at 37 °C. Pierce™ Co-Immunoprecipitation Kit (Thermo Fisher Scientific) was used. In brief, 25 μg of AhR antibody (mouse monoclonal, sc-133088, A-3, Santa Cruz Biotechnology) was covalently coupled to resin for 120 min at room temperature. The antibody-coupled resin was incubated with cell lysate overnight at 4 °C. In parallel with total parental lysates, eluted protein complexes were diluted in delivered sample buffer and resolved on 8% SDS-PAGE gels followed by Western blot analysis and immunodetection with ARNT 1 antibody (mouse monoclonal, sc-17812, G-3, Santa Cruz Biotechnology). Chemiluminescent

detection was performed using horseradish peroxidase-conjugated anti-mouse secondary antibody (7076S, Cell Signaling Technology) and WesternSure® PREMIUM Chemiluminescent Substrate (LI-COR Biotechnology) by a C-DiGit® Blot Scanner (LI-COR Biotechnology). Full scan blots are available at https://doi.org/10.5281/zenodo.7764002.

### Chromatin immunoprecipitation assay

The assay was performed as recently described[47]. Briefly, HepG2 cells were seeded in a 60-mm dish, and the following day, they were incubated with carvones (1000 μM) in combination with vehicle (0.1% DMSO) or the AhR agonists TCDD (20 nM), BaP (7 μM), and FICZ (8 nM) for 90 min at 37 °C. The procedure followed the manufacturer's recommendations for the SimpleChIP Plus Enzymatic Chromatin IP kit (Magnetic Beads) (Cell Signaling Technology; #9005). Anti-AhR rabbit monoclonal antibody was purchased from Cell Signaling Technology (D5S6H; #83200). *CYP1A1* promoter primers were (fw: AGCTAGGC-CATGCCAAAT, rev: AAGGGTCTAGGTCTGCGTGT-3′). Experiments were performed in three consecutive cell passages. Full scan gels are available at https://doi.org/10.5281/zenodo.7764002.

### Protein kinase C inhibition assay

Protein kinase C (PKC) inhibition was assayed in HepaG2 cell lysates using a PKC Kinase Activity Assay Kit (ab139437; Abcam). Cells were grown to 90% confluency in a 60 mm dish. After removal of the medium, 1 mL of lysis buffer (E4030, Promega) was applied for 10 min on ice. Cells were scraped, sonicated, and centrifuged at 15,900 × g/15 min/4 °C (Eppendorf Centrifuge 5415 R; Eppendorf, Stevenage, U.K.). Then, 3 μL of cell lysate was mixed with 297 μL of kinase assay buffer, and 40 μL aliquots were transferred into 0.5 mL microtubes. These aliquots were mixed with 1/100 stock solutions of carvones to obtain final concentrations of 10 μM, 100 μM, and 1000 μM. DMSO (1% V/V) and staurosporine (1 μM) were used as negative and positive controls, respectively. The reaction was initiated by the addition of 10 μL of reconstituted ATP, and the rest of the procedure was performed as described in the manufacturer's recommendations. Absorbance was measured at 450 nm using an Infinite M200 microplate reader (TECAN, Austria). The results are expressed as a percentage of the negative control. The cell lysate was stored at −80 °C and used in performing three independent experiments.

### KINOMEscan™ profiling

The KINOMEscan™ screening platform (scanMAX assay) employs a proprietary active site-directed competition binding assay that quantitatively measures the interactions between test compounds (here 100 μM S-carvone) and 468 human protein kinases[27]. The assay was performed at Eurofins DiscoverX Corp. (Fremont, CA, USA).

### Tyrosine-protein phosphatase non-receptor-type inhibition assays

The catalytic activity of PTPN11/SHP2 and PTPN6/SHP1 was measured with recombinant enzymes incubated with vehicle (DMSO, 0.1% V/V), PTP1B inhibitor (33.3 μM), and S-carvone (100 μM; 1000 μM). The phosphatase activities were monitored as a time-course measurement of the increase in the fluorescence signal from the fluorescent substrate (6,8-difluoro-4-methylumbelliferyl phosphate), and the initial linear portion of the slope (signal/min) was analyzed. Two independent experiments were performed, and the incubations and measurements were performed in duplicate (technical replicates). The assays were carried out at Reaction Biology Corp. (Malvern, PA, USA).

### Heat shock protein 90 kDa fluorescence competitive binding assay

This assay is based on the competition of fluorescently labeled geldanamycin for binding to HSP90. The fluorescent substrate binds to the ATP binding pocket of HSP90; therefore, an ATP-competitive

inhibitor was found by this assay. The assay was carried out at Reaction Biology Corp. (Malvern, PA, USA).

## Molecular modeling and docking

The full-length three-dimensional structure of human AhR has not been resolved. The structure available from alpha fold database consists of several unstructured regions that are unsuitable for understanding the binding mode of carvones. Further, a recent effort by Bourguet´s group has resulted in a high resolution cryo EM structure of indirubin bound HSP90-XAP2-AhR complex[48] but could not be used for understanding the binding mode of carvones as the cryo EM studies failed to resolve the coordinates for the 270 residues from the N-terminal region. The crystal structure complex of a construct of human AhR with a truncated mouse ARNT has been solved (PDB code: 5NJ8)[49]. Since the solved structure does not contain the LBD of AhR, it was modeled based on neuronal PAS-1 protein (PDB code: 5SY5)[47,50]. The molecular structures of carvones were modeled using the ligand builder module of Molecular Operating Environment (MOE ver 2018; Chemical Computing Group; Montreal, Canada). The molecules were energy minimized and geometry optimized for docking studies. Since carvones occupy a small volume and have the potential to bind nearly any binding pocket, we utilized a triage-based approach to finalize the predicted binding pocket. We screened the PAS-B domain of AhR containing the binding pockets for TCDD, FICZ, BaP, CH-223191, vemurafenib, dabrafenib, PLX7904, PLX8394, and resveratrol as detailed in[14] and our newly developed methylindoles[47]. Pockets including TCDD were used as a control for each of these dockings. All docking screening experiments were performed using GOLD version 5.2 (Cambridge Crystallographic Data Centre, Cambridge, UK)[51]. The complexes were ranked using the default option of GOLD SCORE, and the best-ranking complexes were visualized in MOE. The molecules were also docked to AhR derived from the AhR-ARNT complex. S-carvone-bound AhR was then energy minimized and subjected to molecular dynamics simulation with a production run of 10 ns. The docked protein complex of AhR protein with S-carvone was incorporated into an aqueous rectangular box having a dimension of $106 \, nm \times 106 \, nm \times 106 \, nm$. Potassium chloride (0.15 M) was added with extra ions to neutralize the excess charges. The water molecules were modeled as TIP3P water. The initial minimization and equilibration were carried out in our local server using the NAMD software (Version 2.15) and CHARMM36 forcefield. The force field for the ligand was generated using CHARMM General Force Field (CGenFF) program version 2.5.1. The production simulation was carried on the Anton2 supercomputer at the Pittsburgh Supercomputing Center for 400 ns with a 2.5 fs time step. Simulations were run in the NPT ensemble at 310 K and 1 bar using the Nose-Hoover thermostat and the MTK barostat. The cutoff distances for nonbonded interactions were determined automatically by Anton2. Structural snapshots were taken at 10 ns, 100 ns, and 250 ns timepoints and the binding mode of carvone was assessed for the specificity of binding.

## Thermal shift assay

Human AHR 112-272 aa (domain PASA) was subcloned into pMKH vector to produce a his6-TEV-hAHR(PASA) construct. The plasmid was transformed into Rosetta (DE3) cells, and protein was expressed in LB media. Cells were dissolved in lysis buffer containing 20 mM Tris, pH 8.0, 500 mM NaCl, 5% Glycerol, 5 mM imidazole, protease inhibitor cocktail (#5056489001, Sigma Aldrich). Supernatant was collected after sonication and centrifugation, and then flowed through Ni-NTA His-bind resin (#70666-5, Millipore). Resin was washed 3 times with lysis buffer, and protein was eluted with 200 mM imidazole in lysis buffer. The purified protein was passed through a gel filtration column (Cytiva, HiLoad 16/600 Superdex 75) to remove aggregated protein and imidazole. His tag was removed by TEV cleavage and final hAHR(PASA) protein was pooled in size-exclusion chromatography with a Bis-Tris Propane buffer (20 mM Bis-Tris Propane, pH 8.0,

150 mM NaCl). For thermal shift assay, 50 nl of compound was transferred into 384-well plate by Echo 555 liquid handler (Labcyte), and then 5 µl of protein solution was added into each well in a microplate dispenser (#5840300, Thermo Scientific). Protein solution was prepared by diluting hAHR(PASA) protein to 0.1 mg/mL in Bis-Tris Propane buffer, and then add SYPRO orange dye (S6650, Invitrogen) to a final concentration of 8×. Plate was spun at $1000 \times g$ for 10 s and incubated at room temperature for 30 min before transferring into QuantStudio 7 Flex real-time PCR machine (Applied Biosystems). Melt curve was generated by heating the plate from 25 °C to 95 °C applying a gradient of 0.1 °C/s. Data was analyzed in protein thermal shift software v1.4 (Applied Biosystems).

## Cellular thermal shift assay

Human AHR 112–272 aa (domain PASA) was subcloned into pBiT3.1-N vector to produce a HiBiT-hAHR(PASA) construct. HEK293T cells were cultured in DMEM media (#31966-021, Gibco) at 37 °C, 5% $CO_2$. The plasmid was transfected into HEK293T cells with lipofectamine 3000 (L3000001, Invitrogen), and the cells were grown to 60–80% confluence. Cells were harvested 3 days after transfection. Cells were washed two times with ice-cold PBS, scraped and suspended in Bis-Tris Propane buffer containing protease inhibitor cocktail (#5056489001, Sigma). Cells were sonicated and supernatant was collected after centrifugation. The cell lysate was diluted to 0.3 mg/mL for further analysis. 50 nl of compound solution (in DMSO) was transferred into 384-well plate with Echo 555 liquid handler (Labcyte), and 5 µl of cell lysate was added to each well. Plate was spun at $1000 \times g$ for 10 s and incubated at room temperature for 30 min. The plate was heated in PCR thermal cycler (Bio-Rad, C1000 Touch) at a gradient of 38–62 °C for 3 min. Denatured protein was removed by spinning plate at $4300 \times g$ for 15 min, and soluble HiBiT-hAHR(PASA) protein was detected by Nano-Glo HiBiT lytic detection system (N3040, Promega) according to the manufacturer protocol.

## Microscale thermophoresis

A codon-optimized fragment of human AhR (Swiss-Prot database accession number AHR_HUMAN P35869) encoding amino acid residues 23–273 was synthesized and cloned into pET28b(+) using Ndel and BamHI restriction sites to express an N-terminally fused 6×His-tag. A codon-optimized fragment of mouse Arnt encoding amino acid residues 85–345 was synthesized and cloned into pETDuet-1 using BamHI and HindIII restriction sites, expressing N-terminally fused 6×His-tag or using NcoI and HindIII restriction sites, expressing N-terminally FLAG-tag (GenScript, Leiden, Netherlands). A selection of truncated versions of AhR and Arnt was performed based on published data[49,52]. Both constructs were coexpressed in Rosetta 2 (DE3) E. coli cells (Novagen). Protein production was induced with 1 mM isopropyl-$\beta$-thiogalactopyranoside, and cells were grown at 20 °C in LB medium overnight. Cells were lysed at 30 kpsi using a One-Shot cell lyser (Constant Systems Ltd.) and the addition of EDTA-free cOmplete™ protease inhibitor cocktail (Roche). B-PER complete bacterial protein extraction reagent (Thermo) and Denarase (c-LEcta) were added to the lysate. Protein heterodimers were partially purified using HisPur Cobalt columns (Thermo Fisher Scientific) to obtain solutions in the final buffer containing 20 mM HEPES, pH 7.0, 300 mM NaCl, and 5% (w/v) glycerol. The presence of AhR and Arnt proteins was verified by Western blot using anti-His-tag (mouse monoclonal, MA1-21315, dilution 1:1000, Invitrogen) and anti-FLAG-tag (rabbit monoclonal, 14793 S, dilution 1:1000, Cell Signaling Technology) antibodies, respectively. In parallel, lysates from E. coli were separated by electrophoresis using precast NuPAGE Bis-Tris protein gels (Thermo Fisher Scientific) and visualized by Coomassie Brilliant Blue staining. Excised gel pieces with protein bands corresponding to the expected molecular masses of recombinant AhR and Arnt were processed using in-gel digestion and peptide extraction protocols[53], and

the recombinant proteins were identified by nanoflow liquid chromatography of peptides coupled to tandem mass spectrometry[54].

The protein fractions were concentrated to $2\,mg\,mL^{-1}$ using 10 kDa filters (Amicon) and stored at 5 °C for 10 days. Microscale thermophoresis was used to determine S-carvone and D-limonene binding to human 6×His-tagged AhR in a complex with FLAG-Arnt. The protein (200 nM) was fluorescently labeled using a RED-tris-NTA 2nd generation dye (NanoTemper Technologies GmbH) and a 1:1 dye/protein molar ratio in the reaction buffer: 20 mM Tris-HCl, pH 7.4, supplemented with 150 mM NaCl and 0.075% Tween-20. Ligands were dissolved in ethanol (max. 0.5% final concentration in the reaction mixture). Measurements were performed on a Monolith NT.115 instrument (NanoTemper Technologies GmbH) at 25 °C with 3 s/22 s/2 s laser off/on/off times and continuous sample fluorescence recording in premium capillaries and using an excitation power of 90% and a high MST power mode. The normalized fluorescence $\Delta F_{norm}$ [‰] as a function of the ligand concentration was analyzed and concluded to reflect a ligand binding interaction.

In experiments using AhR mutants (Y76A and Y76F), the above-described procedure was applied, using wt-His-hAhR(23-273) plasmid as a template for site-directed mutagenesis (GenScript, Leiden, Netherlands).

## Covalent functionalization of AhR with azido-S-carvone

$N_3$-S-carvone was synthesized according to a published procedure (Fig. S4D)[55]. His-AhR(23-273)/FLAG-Arnt(85-345) were coexpressed in T7 Express *E. coli* cells (Novagen) as described above and reconstituted (0.5 mg/mL) in 20 mM phosphate buffer (pH 7.0) using 10 kDa filters (Amicon). The protein was mixed with 10 mM $N_3$-S-carvone and photoactivated by a 3UV Lamp (Thermo Fisher Scientific) for 1 h at 365 nm and $2\,mW\,cm^{-2}$ intensity. The reaction mixture was resolved by SDS-PAGE, and the gel was stained with QC Colloidal Coomassie S stain (Bio-Rad, Hercules, CA, USA). The protein bands of the recombinant His-AhR(23-273) segment were excised from the gel slab, and their content was subjected to in-gel digestion by SOLu trypsin (Merck, Steinheim, Germany) after a reduction followed by the carbamidomethylation of thiol groups[53]. Peptides from the digests were purified on ZipTip-C18 pipette tips (Merck-Millipore, Carrigtwohill, Ireland) and analyzed by MALDI-TOF/TOF MS and MSMS on an ultrafleXtreme instrument equipped with a Smartbeam II Nd:YAG laser (Bruker Daltonik, Bremen, Germany). Peptide samples (0.5 μL) were deposited on an MTP AnchorChip 384 BC MALDI target (Bruker Daltonik) by a standard dried droplet technique with α-cyano-4-hydroxycinnamic acid matrix (5 mg/mL in 50% acetonitrile containing 0.1% trifluoroacetic acid). The calibration spots on the target were made with Peptide Calibration Standard II (Bruker Daltonik) and the same matrix. The instrumental setups for acquiring mass spectra and tandem mass spectra were as described[54]. MS and MSMS data were processed by flexAnalysis 3.4 and BioTools 3.1 (Bruker Daltonik). Database searches (against the Swiss-Prot protein sequence database) were performed by ProteinScape 3.1 (Bruker Daltonik) and Mascot Server 2.4 (Matrix Science, London, UK) or using PEAKS Studio X (Bioinformatics Solutions, Waterloo, ON, Canada). The mass error tolerances for the MS and MSMS data-based searches were 25 ppm and 0.5 Da, respectively.

## Animal experiments

Six-week-old female C57BL/6 mice were obtained from The Jackson Laboratory (Bar Harbor, ME, USA; #000664) and housed for two weeks in the institutional vivarium before experimentation. They were co-housed for acclimatization at the vivarium for one weeks prior to experiments. Housing conditions: 14 h light/ 10 h dark cycle; temperature: 20–22 °C; humidity: 30–70%; diet: LAB Diet #5058. All clinical inspections were performed by laboratory personnel. On our animal protocol, mice were clinically inspected daily (with particular attention to ear and body skin conditions at 6–8 h intervals within a single day). For irradiation studies, there is potential for ear bleeding, ulceration, and infection. These could result in poor movement or feeding although this was not observed in any mice over a 24 h period of observation. Mice were to be euthanized if they exhibited–poor feeding, ulcerated skin on ears, cachexia, weight loss >20% of highest basal weight, and poor drinking–mice were to be hydrated for loss of fluids (50–100 μL in 0.9% saline/PBS every 3–4 h to look for signs of reversal), shallow breathing. None of the mice in the study met any criteria for euthanasia. At the end of the experiment, all mice were euthanized by $CO_2$ asphyxiation.

The experiments were approved by the Institutional Animal Care and Use Committee of the Albert Einstein College of Medicine (New York, NY, USA; Protocol #00001405). They were performed with the following institutional and national guidelines[56]. Since all mice are inbred, active, and had nearly equivalent starting weights/overall body habitus, mice were randomly picked without prior knowledge of baseline weight from each cage and assigned to control versus treatment group(s) in a consecutive manner. No randomization software was used. Since this is an exploratory analysis to generate the hypothesis that carvones protect against AhR mediated UV-damage as an in vivo "proof-of-concept" for its antagonist actions on ligand-activated AhR, a priori sample size calculations for the treatment groups and controls were not conducted. Instead, given the technical difficulty of managing more than 3 mice per treatment group for the UV studies, we performed all the experimental groups in two installments spread over time using $n = 3/$ treatment group. Thus, in total, we obtained $n = 6$ mice/treatment, which was included in the analyses. Experiments with RT-qPCR analysis endpoints were performed separately from experiments with H&E endpoints. No mice died or had sickness to preclude and replace the sample. Mouse ears were irradiated with short-wave UV 254 nm light (distance: ~1 cm; absorbed dose: $360\,mJ/cm^2$) using a Spectronics ENF-240C Handheld UV Lamp (Spectronics Corp., Melville, NY, USA). Chemicals were dissolved in acetone and topically applied to the skin (mouse ear). (i) BaP (1 μg) was applied in two doses (0 h, 16 h); (ii) S-carvone (960 μg) was applied 1 h before the first BaP dose; (iii) UV irradiation was applied at 0 h and −1 h. In each mouse, the left and right ears were exposed to different treatments, thereby providing internal individual controls for comparative treatments, and each animal then represents a biological repeat. The ears were collected at 24 h, and RNA was isolated using TRI Reagent® (Carlsbad, CA, USA, #15596026). Note: In these experiments, mouse positioning relative to the UV lamp is critical to get even exposures across entire ear, so this was optimized individually for each mouse. Adjustments of distance from the source and time of exposure will also need to be optimized for a given mouse since that natural ear positioning and curvature is different from each mouse.

**Ear histology & immunohistochemistry.** Following auriculectomy, the ear tissue was rinsed, fixed in 10% buffered formalin for 24 h, and embedded in paraffin. Slices (3 mm) were cut and, for sun (UV)-burned cell scoring purposes, stained with hematoxylin-eosin. The sunburn cell count was performed by HL in a single-blinded manner; the slides were decoded only after the cell count was determined. Immunohistochemical staining against gamma-H2AX-Ser139-P was performed using P-Histone H2AX primary antibody (Cell Signaling, CST9718, 1:800). Bond Polymer Refine Detection (Leica Biosystems) was used according to the manufacturer's protocol. After staining, sections were dehydrated, and film cover-slipped using a TissueTek-Prisma and Coverslipper (Sakura). Whole-slide scanning (40×) was performed on an Aperio AT2 (Leica Biosystems). Immunohistochemistry was carried out at HistoWiz, Inc. (New York City, NY, USA). Note: In these studies, ear embedding needs to be adjusted and the cut level appropriate to obtain an accurate representation of the entire "continuous" auricular epithelium. This was optimized for each mouse ear through repeated embedding and cuts adjusted to obtain a continuous epithelial layer.

## Statistics

All statistical analyses, as well as the calculations of half-maximal effective concentration ($EC_{50}$), $EC_{80}$, and half-maximal inhibitory concentration ($IC_{50}$) values, were performed using GraphPad Prism 8 for Windows (GraphPad Software, La Jolla, CA, USA).

The numbers of independent repeats and technical replicates are stated in the corresponding figure legends for all the experiments. Where appropriate, data were processed by one-way analysis of variance (ANOVA) followed by Dunnett's test or Student's t test. The results with $p$ values lower than 0.05 were considered significant. The $EC_{50}$, $EC_{80}$, and $IC_{50}$ values were calculated using nonlinear regression by the least-square fitting method. The R-squared value was checked in all of the calculations and did not drop below 0.9. The inhibition constant ($K_i$) was calculated using the Cheng-Prusoff equation[57]. In an animal experiment, the normality of the data was analyzed by Shapiro-Wilk test, the outliers were detected by Grubbs test, and the significance was determined by Mann-Whitney non-parametric test.

## Reporting summary

Further information on research design is available in the Nature Portfolio Reporting Summary linked to this article.

## Data availability

All data needed to evaluate the paper's conclusion are presented in the paper or the Supplementary Materials. Source data are available with this manuscript and have also been deposited in a publicly accessible repository: https://doi.org/10.5281/zenodo.7764002. Source data are provided in this paper.

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

## Acknowledgements

Financial support from the Czech Health Research Council [NV19-05-00220] (to Z.D.), the Czech Science Foundation [23-04662S] (to Z.D.), the National research council at the National Academies of Science and Pittsburgh supercomputing facility for use of Anton2 with grant [MCB210021P] (to S.K), and the Juergen Manchot Foundation (to K.M.R.) are acknowledged. We thank Dr. Radka Končitíková for assistance with the microscale thermophoresis experiments and Dr. David Vanda for support with the synthesis of azidocarvone.

## Author contributions

Participated in research design: Z.D., T.H.S., S.M. Conducted experiments: K.O., B.V., I.Z., K.K., E.M., R.V., P.N., K.M.R., S.K., D.K., M.K., M.Š., H.L., M.S., H.P., B.N. Contributed new reagents and analytic tools: Z.D., T.H.S., S.K. Performed data analysis: K.O., B.V., I.Z., K.K., E.M., R.V., P.N., K.M.R., S.K., Z.D., D.K., M.K., M.Š., H.L., H.P., F.R., B.N. Wrote or contributed to the writing of the manuscript: Z.D., T.H.S., S.M., S.K.

## Competing interests

The authors declare no competing interests.
