## [Peer Review File · Nature Communications]

MONOTERPENOID ARYL HYDROCARBON RECEPTOR
ALLOSTERIC ANTAGONISTS PROTECT AGAINST ULTRAVIOLET
SKIN DAMAGE IN FEMALE MICEReviewers' Comments:

Reviewer #1:

Remarks to the Author:

The manuscript entitled "Aryl hydrocarbon receptor allosteric antagonists: a new paradigm established by monoterpenoids" reports on the discovery of carvones, which are monoterpenoids present in specific essential oils, as allosteric modulators of the human aryl hydrocarbon receptor (AhR). Indeed, several compounds have been described as AhR modulators but the authors' central claim is that carvones are the first to be characterized as noncompetitive modulators. To support this finding, the authors convincingly evidence the noncompetitive MoA and try to give hints on the possible binding site on the AhR. In particular, docking experiments, microscale thermophoresis, and MS experiments allow inferring a possible binding site. Unfortunately, these results are not conclusive although plausible. I would suggest the authors further elaborate on this part of the work by performing site-directed mutagenesis experiments on the residues indicated as being critical for ligand binding followed by microscale thermophoresis. Also, I would suggest the authors performed preliminary structure-activity relationship studies to show how ligand modifications affect the binding/activity of carvones.

From the technical point of view, the authors should better describe the methods employed to perform the MD simulations. Also, 10 ns is a very short sampling time to convincingly show that the detected protein fluctuations are not transient.

I agree with the authors that allosteric modulation might have advantages over the orthosteric one. Nevertheless, no experimental data were provided to substantiate that this applies to the case of AhR modulation. Indeed a comparison of the biological effect of carvones and an orthosteric antagonist should be provided. The same ligand should be used as a control in all the experiments attained to show the noncompetitive MoA of carvones.

Reviewer #2:

Remarks to the Author:

This manuscript by Poulíková details the characterization of carvones as non-competitive inhibitors of the aryl hydrocarbon receptor (AhR). While agonists of AhR have been well-characterized, antagonists are rare and of potential use in a variety of both basic science and therapeutic settings. The authors show strong evidence for this effect on transcription of both reporters and endogenous AhR-driven genes. Similarly strong co-IP data and promoter occupancy data suggest that these antagonists block the essential AhR:ARNT heterodimerization and subsequent DNA binding steps. In vitro data from microscale thermophoresis and mass spectrometry are combined with molecular docking to suggest a binding site, but I have substantial concerns about these data as noted below. A number of possible off-target mechanisms of carvones are checked and eliminated.

From my perspective, the cellular data and initial mechanistic investigations are interesting and data are chiefly compelling (note minor concern about nuclear translocation, below). However, the Fig 4 data that is central to testing the mode of action have significant flaws in my opinion, and make this work very preliminary at this time. I encourage the authors to address these concerns, and further, to test the impact of point mutations on proposed binding sites to see if these disrupt binding.

Major concerns:

- a major concern of mine is the molecular docking and in vitro biochemistry experiments using of truncated constructs shown in Fig 4. The docking is done on a isolated AhR PASB model, rather than integrating such into larger bHLH-PASA-PASB models; the biochemistry is done on a bHLH-PASA heterodimer. Such constructs ignore the absolutely-critical PASB domains that bind AhR agonists and establish another set of AhR:ARNT protein-protein interactions, such as seen in other bHLH-PAS transcription factors. While I am aware that there are technical challenges in this, they are

surmountable – generate an AlphaFold2 model of AhR:ARNT for the docking and evaluate other constructs for ligand binding, for example. While I appreciate that the data in Fig 4C suggest some binding of carvones to this subfragment, the affinities here are much worse (~100x) than the cell-based data, raising substantial concerns about the validity of these data to show a specific binding event.

- mass spec data suffer from the same concern raised above with truncated AhR fragments, raising substantial concerns about their relevance.

Minor concerns:

- the authors routinely use n=2 replicates for cellular assays, such as those shown in Figs 1 and 2. This is not ideal compared to triplicates; different journals have different requirements in this area, and I leave it to the editor to establish if it is acceptable here.

- nuclear localization data shown in Fig 3A are not particularly compelling, particularly in the expected increase in nuclear localization between vehicle and the three AhR agonists.

Reviewer #3:

Remarks to the Author:

Carvones have previously been published as weak AHR antagonist, so the potential novelty of this work is the concept of non-competitive antagonism of AHR activity at a site other than the ligand binding pocket. Considering that the AHR appears to have hydrophobic patches on the surface the concept put forward in this article is reasonable. However, considering the 1 mM concentration needed to alter ligand binding in vitro or the 1 mM dose needed to inhibit AHR mediated Cyp1a1 induction, the physiological significance of this effect has not been demonstrated. Additional control compounds need to be utilized to establish the specificity of the carvone antagonism through essentially all the experiments. Also, throughout all cell culture experiments cell viability assessment should be performed. Perhaps the single most important point is the authors have failed to make a convincing case that carvones bind to a site other than the actual ligand binding pocket.

Specific comments:

1. The reporter experiment to examine antagonism was performed at 24 h, at this time point considerable metabolism occurs. With this in mind, the ability of 10 uM carvones to inhibit TCDD is surprising as TCDD is not metabolized. In these experiments there is no assessment of toxicity in cells performed, a required control for these experiments.

2. The dose that inhibits AHR activity in the reporter experiments appears to be lower than what is observed in Cyp1a1 mRNA induction experiments, this could be due to interference with luciferase. Some compounds have been shown to inhibit luciferase activity.

3. In figure 4 carvones only inhibit in the binding assay at 1 mM, this result could be due to non-specific binding, and it is critical that control with similar compounds are utilized in the assay. Does D-limonene inhibit in this assay at 1 mM?

4. The only actual data for making the statement that carvones bind to a different site on the AHR is not convincing and is based on a model of the AHR structure. The CryoEM structure has just been published in preprint form and perhaps this could enhance the modeling.

5. The generation of the azide form of carvones could be used to examine where the covalent binding

occurs in the peptide structure, like what has been performed by Alan Poland's laboratory with a dioxin derivative.

6. In figure S4 the cartoon in panel A gives the impression that D-limonene does not inhibit AHR activity, yet the data in panel B shows a statistically significant repression.

7. The data in figure 6 could be due to a sunscreen effect of carvones on UV damage. This experiment lacks the appropriate control treating with carvones just after UV exposure. In addition, the use of Ahr null mice would be required to interpret this data.

RESPONSE TO THE REVIEWERS

REVIEWER #1

The manuscript entitled “Aryl hydrocarbon receptor allosteric antagonists: a new paradigm established by monoterpenoids” reports on the discovery of carvones, which are monoterpenoids present in specific essential oils, as allosteric modulators of the human aryl hydrocarbon receptor (AhR). Indeed, several compounds have been described as AhR modulators but the authors’ central claim is that carvones are the first to be characterized as noncompetitive modulators. To support this finding, the authors convincingly evidence the noncompetitive MoA and try to give hints on the possible binding site on the AhR. In particular, docking experiments, microscale thermophoresis, and MS experiments allow inferring a possible binding site.

We thank the reviewer for valuable comments, which we addressed as follows:

Comment 1/1

Unfortunately, these results are not conclusive although plausible. I would suggest the authors further elaborate on this part of the work by performing site-directed mutagenesis experiments on the residues indicated as being critical for ligand binding followed by microscale thermophoresis.

Response 1/1

We performed microscale thermophoresis using the AhR mutants, Y76F and Y76A (a residue we have shown critical to binding). We show that carvones do not bind to Y76A mutant and that carvones bind Y76F mutant much weaker as compared to wt-AhR. These data provide the necessary evidence that residue Y76 on AhR is indeed a carvone-interaction (binding) site. The data are presented in Figure 4D in the revised manuscript. In addition, we have performed thermal shift analyses with purified and cellular human PAS-A domain (residues 112-272), which showed that carvones do not interact with AhR PAS-A domain, therefore, providing additional evidence that critical binding site for carvones should be located within 23-111 region of human AhR.

Comment 1/2

Also, I would suggest the authors performed preliminary structure-activity relationship studies to show how ligand modifications affect the binding/activity of carvones.

Response 1/2

We have performed a SAR for monoterpenoids (n = 14) that included reporter assays and MTT assay as the end-points of assessment. Additionally, for monoterpenoids with significant AhR antagonist activity (n = 7), we carried out protein co-immune precipitation (co-IP; AHR:ARNT) and AhR nuclear localization studies. These data are exhaustive and they are presented in Figure S2, Figure S5A and Figure S5B in the revised manuscript.

Comment 1/3

From the technical point of view, the authors should better describe the methods employed to perform the MD simulations. Also, 10 ns is a very short sampling time to convincingly show that the detected protein fluctuations are not transient.

Response 1/3

(i) We have now included an additional description for the modeling and MD simulations sections in the manuscript. We agree that 10 ns was a short sampling time and hence extended the simulations to 400 ns. We performed MD simulations using the increased length of the simulation and completed 400 ns long simulations. Supplementary Figure S7 details the comparison between 10 ns complex of Ahr-S-carvone with that of the 250 ns complex. While S-carvone is a small molecule docked to a fairly large binding pocket, the interactions within

the binding pocket at 10 ns and 250 ns remain conserved suggesting that the pocket may be specific for S-carvones. Based on these simulation results, we proposed to mutate Y76 a residue that is within 3.5Å of carvone during the entire simulation to validate the binding specificity of carvone.

(ii) We reviewed the full-length structure of AhR using alphafold2 as suggested by reviewer 2, however, the structure had several unstructured regions, which were not useful for understanding the dimeric forms. A structural superposition of our model at 100 ns and 400 ns onto the alpha fold AhR structure shows the overall conservation of the secondary structural elements and binding pocket for S-carvones as shown in the figure.

According to the suggestion provided by reviewer 3 to use cryoEM AhR:Hsp90 XAP2 structure, we have obtained the unpublished coordinates of the complex from the authors (pre-print posted at BioRxives: <https://doi.org/10.1101/2022.05.17.491947>). However, the cryoEM structure of AhR derived from the AhR:Hsp90 XAP2 complex lacks residues 1-270; a domain, which we have demonstrated to be involved in binding carvones. Hence it was not used for docking carvones.

Comment 1/4

I agree with the authors that allosteric modulation might have advantages over the orthosteric one. Nevertheless, no experimental data were provided to substantiate that this applies to the case of AhR modulation. Indeed, a comparison of the biological effect of carvones and an orthosteric antagonist should be provided. The same ligand should be used as a control in all the experiments attained to show the noncompetitive MoA of carvones.

Response 1/4

We have performed additional reporter gene experiments with CH223191, a classical orthosteric antagonist of AhR. Exhaustive analyses were carried out, including 4 h and 24 h treatments with CH223191, by combining with a series of model AhR agonists (n = 6). The data are presented in Figure 1 in the revised manuscript.

However, we do not feel that we should repeat all existing studies performed with carvones adding on CH223191 in the experimental design. This is obviated by the data we already show on CH223191 in reporter gene assays.

REVIEWER #2

This manuscript by Poulíková details the characterization of carvones as non-competitive inhibitors of the aryl hydrocarbon receptor (AhR). While agonists of AhR have been well-characterized, antagonists are rare and of potential use in a variety of both basic science and therapeutic settings. The authors show strong evidence for this effect on transcription of both reporters and endogenous AhR-driven genes. Similarly strong co-IP data and promoter occupancy data suggest that these antagonists block the essential AhR:ARNT heterodimerization and subsequent DNA binding steps. In vitro data from microscale thermophoresis and mass spectrometry are combined with molecular docking to suggest a binding site, but I have substantial concerns about these data as noted below. A number of possible off-target mechanisms of carvones are checked and eliminated. From my perspective, the cellular data and initial mechanistic investigations are interesting and data are chiefly compelling (note minor concern about nuclear translocation, below).

We thank the reviewer for valuable comments, which we addressed as follows:

Comment 2/1

However, the Fig 4 data that is central to testing the mode of action have significant flaws in my opinion, and make this work very preliminary at this time. I encourage the authors to address these concerns, and further, to test the impact of point mutations on proposed binding sites to see if these disrupt binding.

Response 2/1

Please refer to the Response 1/1 to the Reviewer #1.

Comment 2/2

a major concern of mine is the molecular docking and in vitro biochemistry experiments using of truncated constructs shown in Fig 4. The docking is done on a isolated AhR PASB model, rather than integrating such into larger bHLH-PASA-PASB models; the biochemistry is done on a bHLH-PASA heterodimer. Such constructs ignore the absolutely-critical PASB domains that bind AhR agonists and establish another set of AhR:ARNT protein-protein interactions, such as seen in other bHLH-PAS transcription factors. While I am aware that there are technical challenges in this, they are surmountable – generate an AlphaFold2 model of AhR:ARNT for the docking and evaluate other constructs for ligand binding, for example. While I appreciate that the data in Fig 4C suggest some binding of carvones to this subfragment, the affinities here are much worse (~100x) than the cell-based data, raising substantial concerns about the validity of these data to show a specific binding event.

Response 2/2

Please refer to the Response 1/1 and the Response 1/3 to the Reviewer #1.

Comment 2/3

mass spec data suffer from the same concern raised above with truncated AhR fragments, raising substantial concerns about their relevance.

Response 2/3

Please refer to the Response 2/2 to the Reviewer #2.

Comment 2/4

the authors routinely use n=2 replicates for cellular assays, such as those shown in Figs 1 and 2. This is not ideal compared to triplicates; different journals have different requirements in this area, and I leave it to the editor to establish if it is acceptable here.

Response 2/4

In the revised paper, we have only used n = 2 technical replicates for screening assays as reported in Fig 5 (off-target effects) and Table S3 (to show nuclear translocation). These data are not the main data and show clear consistency between the technical replicates. There were at least 2 independent biologic experiments conducted for each and for Table S3 the experiment was done in two consecutive cell passages. We do not feel that performing another replicate will change our conclusions.

Comment 2/5

nuclear localization data shown in Fig 3A are not particularly compelling, particularly in the expected increase in nuclear localization between vehicle and the three AhR agonists.

Response 2/5

The Fig 3A has been quantified in Table S3. The data is clear as there is a > 10-fold increase in nuclear localized signals with all 3 ligands across the 2 independent biologic experiments (each performed with technical duplicates).

REVIEWER #3

Carvones have previously been published as weak AHR antagonist, so the potential novelty of this work is the concept of non-competitive antagonism of AHR activity at a site other than the ligand binding pocket. Considering that the AHR appears to have hydrophobic patches on the surface the concept put forward in this article is reasonable.

We thank the reviewer for valuable comments, which we addressed as follows:

Comment 3/1

However, considering the 1 mM concentration needed to alter ligand binding in vitro or the 1 mM dose needed to inhibit AHR mediated Cyp1a1 induction, the physiological significance of this effect has not been demonstrated.

Response 3/1

The predicted range of concentration is based on our application of 1 mg carvone to an entire mouse ear. If we do a w/v approximation (volume from complete homogenization), the approximated volume of a mouse ear ~ 30-50 microL. Thus, the concentration of carvones is at least in the range of 200 mM.

Comment 3/2

Additional control compounds need to be utilized to establish the specificity of the carvone antagonism through essentially all the experiments.

Response 3/2

Please refer to the Response 1/4 to the Reviewer #1.

Comment 3/3

Also, throughout all cell culture experiments cell viability assessment should be performed.

Response 3/3

Cell viability assay (MTT) was performed, and carvones are not cytotoxic in the concentration used in the study. The data are presented in Figure S1E in the revised manuscript.

Comment 3/4

Perhaps the single most important point is the authors have failed to make a convincing case that carvones bind to a site other than the actual ligand binding pocket.

Response 3/4

Please refer to the Response 1/1 to the Reviewer #1.

Comment 3/5

The reporter experiment to examine antagonism was performed at 24 h, at this time point considerable metabolism occurs. With this in mind, the ability of 10 uM carvones to inhibit TCDD is surprising as TCDD is not metabolized. In these experiments there is no assessment of toxicity in cells performed, a required control for these experiments.

Response 3/5

The outcome of cell-based antagonist assay depends on multiple variables, including transmembrane traffic and metabolism of both agonist and antagonist etc. Therefore, we used co-treatments at one time, and used incubation times 4 h and 24 h. Please refer to Fig 1 and Fig S1. The short incubation at 4 h shows similar trends as at 24 h. For cytotoxicity – see Response 3/3 above.

Comment 3/6

The dose that inhibits AHR activity in the reporter experiments appears to be lower than what is observed in Cyp1a1 mRNA induction experiments, this could be due to interference with luciferase. Some compounds have been shown to inhibit luciferase activity.

Response 3/6

Carvone does not inhibit luciferase catalytic activity. The data are presented in Figure S1D in the revised manuscript.

Comment 3/7

In figure 4 carvones only inhibit in the binding assay at 1 mM, this result could be due to non-specific binding, and it is critical that control with similar compounds are utilized in the assay. Does D-limonene inhibit in this assay at 1 mM?

Response 3/7

We performed an additional experiment and did ligand binding assays with D-limonene. Indeed, non-specific binding was observed also for D-limonene. The data are presented in Figure S6D in the revised manuscript.

Comment 3/8

The only actual data for making the statement that carvones bind to a different site on the AHR is not convincing and is based on a model of the AHR structure. The CryoEM structure has just been published in preprint form and perhaps this could enhance the modeling.

Response 3/8

Please refer to the Response 1/3 to the Reviewer #1.

Comment 3/9

The generation of the azide form of carvones could be used to examine where the covalent binding occurs in the peptide structure, like what has been performed by Alan Poland's laboratory with a dioxin derivative.

Response 3/9

To be consistent and compatible with microscale thermophoresis experiments, we used the same AhR truncated fragment for chemical modification with azide-carvone. Thus, we are not sure what the reviewer is requesting us to do here that will shed light on the binding of carvones to specific residues.

Comment 3/10

In figure S4 the cartoon in panel A gives the impression that D-limonene does not inhibit AHR activity, yet the data in panel B shows a statistically significant repression.

Response 3/10

Yes this is correct – the extent of inhibition by D-limonene is < 5% but with the number of experimental replicates we find statistical significance, but this does not reflect biological significance. On the other hand, carvones show > 80% inhibition regardless of ligand which is also statistically significant but clearly has biological significance.

Comment 3/11

The data in figure 6 could be due to a sunscreen effect of carvones on UV damage. This experiment lacks the appropriate control treating with carvones just after UV exposure. In addition, the use of Ahr null mice would be required to interpret this data.

Response 3/11

We performed additional animal experiments, and we added the results showing the effects of UV vs UV+carvone. Carvone did not affect the expression of cyp1A1 and cxcl5 mRNAs

under UV-exposure. In addition, carvone itself did not influence the number of UV-burned cells, regardless of its application prior or post UV-exposure. These data disprove the potential shielding effect of S-carvone against UV-irradiation.

Since the effect of Carvones is postulated to occur *via* activated Ahr antagonism, loss of Ahr will not prove or disprove this effect since there is no Ahr to activate. Thus, we feel in this situation, adding Ahr KO mice will not yield additional important insight into *in vivo* mechanisms. The data are presented in Figure 6 in the revised manuscript.

Reviewers' Comments:

Reviewer #1:

Remarks to the Author:

The authors have successfully addressed my concerns by performing most of the requested additional experiments further substantiating their finding. I am now endorsing the manuscript publication in the Nature Communications Journal in the revised form.

Reviewer #2:

Remarks to the Author:

This revised manuscript by Ondrová and coworkers details the characterization of carvones as non-competitive inhibitors of the aryl hydrocarbon receptor (AhR). As noted before, the strength of this manuscript is multifaceted evidence for this inhibitory effect on transcription of both reporters and endogenous AhR-driven genes, complemented by mechanistic investigation that suggest the carvones can block the essential AhR:ARNT heterodimerization and subsequent DNA binding steps.

In revision, the authors have bolstered prior microscale thermophoresis data with a newly presented comparison of carvone binding to a AhR (WT, Y76A, Y76F):ARNT bHLH-PASA heterodimer is a key addition.

While I remain to have some lingering hesitation on aspects of this manuscript due to the strength/replicates of certain experiments as well as some of the authors' choices in presenting their data, I believe that the strength of the novelty and breadth of these findings warrant their publication. Some minor points can be trivially addressed in revision:

Minor concerns;

- the figures would benefit from a review, and potentially, redesign in certain cases. Key items include: several figures are presented without units on axes (Fig 1, where all log c are presumably molar) or incorrect ones (Fig S1B, where log c is definitely not micromolar), and changed axis ranges across time/compound titrations where use of common sets would facilitate comparisons. Graphs were also complicated by black and white graphics, when color could aid readers' analyses of plots with multiple datasets superimposed. I encourage a thorough review.

- consider adding a structure of CH223191 to Fig 1

Reviewer #3:

Remarks to the Author:

The authors have addressed many of the technical issues and necessary new experiments raised in the first review, the data now provides firm support for a novel allosteric mechanism of carvone inhibition of AHR/ARNT heterodimerization. Considering the concentration need to mediate a significant inhibition of AHR transcriptional activity the physiological relevance of the observations is difficult to see. The value in this work could be to serve as a framework for the development of more potent allosteric inhibitors of pharmacological value. However, considering the number of high affinity AHR antagonist already available the importance of this approach is uncertain. Reviewers' Comments:

Reviewer #1:

Remarks to the Author:

The authors have successfully addressed my concerns by performing most of the requested additional experiments further substantiating their finding. I am now endorsing the manuscript publication in the Nature Communications Journal in the revised form.

Reviewer #2:

Remarks to the Author:

This revised manuscript by Ondrová and coworkers details the characterization of carvones as non-competitive inhibitors of the aryl hydrocarbon receptor (AhR). As noted before, the strength of this manuscript is multifaceted evidence for this inhibitory effect on transcription of both reporters and endogenous AhR-driven genes, complemented by mechanistic investigation that suggest the carvones can block the essential AhR:ARNT heterodimerization and subsequent DNA binding steps.

In revision, the authors have bolstered prior microscale thermophoresis data with a newly presented comparison of carvone binding to a AhR (WT, Y76A, Y76F):ARNT bHLH-PASA heterodimer is a key addition.

While I remain to have some lingering hesitation on aspects of this manuscript due to the strength/replicates of certain experiments as well as some of the authors' choices in presenting their data, I believe that the strength of the novelty and breadth of these findings warrant their publication. Some minor points can be trivially addressed in revision:

Minor concerns;

- the figures would benefit from a review, and potentially, redesign in certain cases. Key items include: several figures are presented without units on axes (Fig 1, where all log c are presumably molar) or incorrect ones (Fig S1B, where log c is definitely not micromolar), and changed axis ranges across time/compound titrations where use of common sets would facilitate comparisons. Graphs were also complicated by black and white graphics, when color could aid readers' analyses of plots with multiple datasets superimposed. I encourage a thorough review.

- consider adding a structure of CH223191 to Fig 1

Reviewer #3:

Remarks to the Author:

The authors have addressed many of the technical issues and necessary new experiments raised in the first review, the data now provides firm support for a novel allosteric mechanism of carvone inhibition of AHR/ARNT heterodimerization. Considering the concentration need to mediate a significant inhibition of AHR transcriptional activity the physiological relevance of the observations is difficult to see. The value in this work could be to serve as a framework for the development of more potent allosteric inhibitors of pharmacological value. However, considering the number of high affinity AHR antagonist already available the importance of this approach is uncertain.

RESPONSE TO THE REVIEWERS

REVIEWER #1

The authors have successfully addressed my concerns by performing most of the requested additional experiments further substantiating their finding. I am now endorsing the manuscript publication in the Nature Communications Journal in the revised form.

RESPONSE: We thank the reviewer for this positive assessment.

REVIEWER #2

This revised manuscript by Ondrová and coworkers details the characterization of carvones as non-competitive inhibitors of the aryl hydrocarbon receptor (AhR). As noted before, the strength of this manuscript is multifaceted evidence for this inhibitory effect on transcription of both reporters and endogenous AhR-driven genes, complemented by mechanistic investigation that suggest the carvones can block the essential AhR:ARNT heterodimerization and subsequent DNA binding steps. In revision, the authors have bolstered prior microscale thermophoresis data with a newly presented comparison of carvone binding to a AhR (WT, Y76A, Y76F):ARNT bHLH-PASA heterodimer is a key addition. While I remain to have some lingering hesitation on aspects of this manuscript due to the strength/replicates of certain experiments as well as some of the authors' choices in presenting their data, I believe that the strength of the novelty and breadth of these findings warrant their publication. Some minor points can be trivially addressed in revision:

Minor concerns;

- the figures would benefit from a review, and potentially, redesign in certain cases. Key items include: several figures are presented without units on axes (Fig 1, where all log c are presumably molar) or incorrect ones (Fig S1B, where log c is definitely not micromolar), and changed axis ranges across time/compound titrations where use of common sets would facilitate comparisons. Graphs were also complicated by black and white graphics, when color could aid readers' analyses of plots with multiple datasets superimposed. I encourage a thorough review.

- consider adding a structure of CH223191 to Fig 1

RESPONSE: We thank the reviewer for this positive assessment.

- all figures has undergone extensive review, also according to the editorial comments and journal policy

- a structure of CH223191 was included in the revised Figure 1

REVIEWER #3

The authors have addressed many of the technical issues and necessary new experiments raised in the first review, the data now provides firm support for a novel allosteric mechanism of carvone inhibition of AHR/ARNT heterodimerization. Considering the concentration need to mediate a significant inhibition of AHR transcriptional activity the physiological relevance of the observations is difficult to see. The value in this work could be to serve as a framework for the development of more potent allosteric inhibitors of pharmacological value. However, considering the number of high affinity AHR antagonist already available the importance of this approach is uncertain.

RESPONSE: We thank the reviewer for this positive assessment. Indeed, we are aware of the relatively high concentration of carvone need to mediate a significant inhibition of AHR transcriptional activity. We already carried out large-scale SAR screen of existing natural monoterpeneoids (some of them presented in the revised paper), which served as a base for developing new synthetic and more potent derivatives. Whereas the gross affinity of carvone is much lower than that of existing orthosteric AHR antagonists, the novelty and the potential of our data are represented by the identification of functional allosteric binding site for carvone.